# Improved Distribution Matching Distillation for Fast Image Synthesis

Tianwei Yin[1]     Michaël Gharbi[2]     Taesung Park[2]     Richard Zhang[2]
Eli Shechtman[2]     Frédo Durand[1]     William T. Freeman[1]

[1]Massachusetts Institute of Technology     [2]Adobe Research

https://tianweiy.github.io/dmd2/

## Abstract

Recent approaches have shown promises distilling expensive diffusion models into efficient one-step generators. Amongst them, Distribution Matching Distillation (DMD) produces one-step generators that match their teacher in *distribution*, i.e., the distillation process does not enforce a one-to-one correspondence with the sampling trajectories of their teachers. However, to ensure stable training in practice, DMD requires an additional regression loss computed using a large set of noise–image pairs, generated by the teacher with many steps of a deterministic sampler. This is not only computationally expensive for large-scale text-to-image synthesis, but it also limits the student's quality, tying it too closely to the teacher's original sampling paths. We introduce DMD2, a set of techniques that lift this limitation and improve DMD training. First, we eliminate the regression loss and the need for expensive dataset construction. We show that the resulting instability is due to the "fake" critic not estimating the distribution of generated samples with sufficient accuracy and propose a two time-scale update rule as a remedy. Second, we integrate a GAN loss into the distillation procedure, discriminating between generated samples and real images. This lets us train the student model on *real* data, thus mitigating the imperfect "real" score estimation from the teacher model, and thereby enhancing quality. Third, we introduce a new training procedure that enables multi-step sampling in the student, and addresses the training–inference input mismatch of previous work, by simulating inference-time generator samples during training. Taken together, our improvements set new benchmarks in one-step image generation, with FID scores of 1.28 on ImageNet-64×64 and 8.35 on zero-shot COCO 2014, surpassing the original teacher despite a $500\times$ reduction in inference cost. Further, we show our approach can generate megapixel images by distilling SDXL, demonstrating exceptional visual quality among few-step methods, and surpassing the teacher. We release our code and pretrained models.

## 1   Introduction

Diffusion models have achieved unprecedented quality in visual generation tasks [1–8]. But their sampling procedure typically requires dozens of iterative denoising steps, each of which is a forward pass through a neural network. This makes high resolution text-to-image synthesis slow and expensive. To address this issue, numerous distillation methods have been developed to convert a teacher diffusion model into an efficient, few-step student generator [9–20]. However, they often result in degraded quality, as the student model is typically trained with a loss to learn the pairwise noise-to-image mapping of the teacher, but struggles to perfectly mimic its behavior.

38th Conference on Neural Information Processing Systems (NeurIPS 2024).

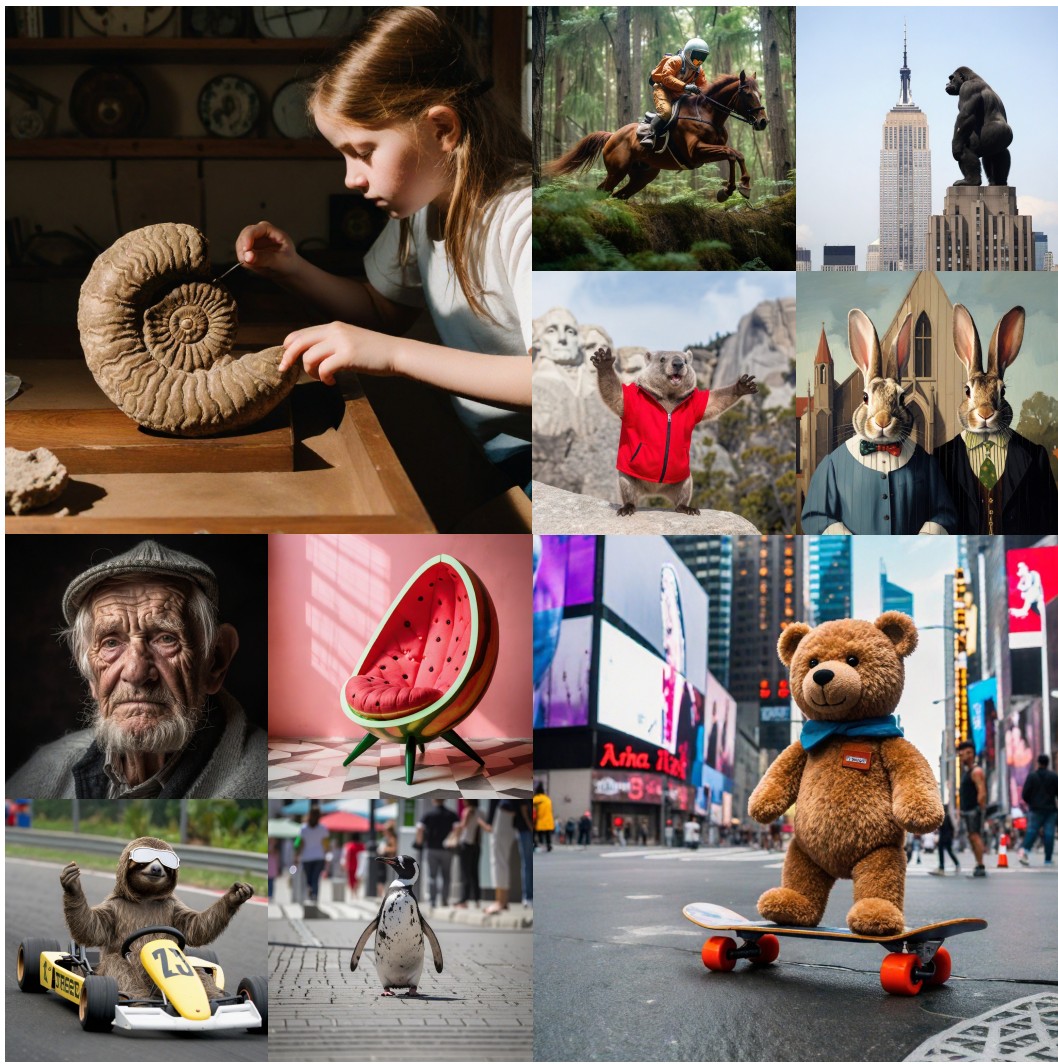

Figure 1: 1024×1024 samples produced by our 4-step generator distilled from SDXL. Please zoom in for details.

Nevertheless, it should be noted that loss functions aimed at matching distributions, such as the GAN [21] or the DMD [22] loss, are not burdened with the complexity of precisely learning the specific paths from noise to image because their goal is to align with the teacher model in terms of *distribution*—by minimizing either a Jensen-Shannon (JS) or an approximate Kullback-Leibler (KL) divergence between the student and teacher output distributions.

In particular, DMD [22] has demonstrated state-of-the-art results in distilling Stable Diffusion 1.5, yet it remains less investigated than GAN-based methods [23–29]. A likely reason is that DMD still requires an additional regression loss to ensure stable training. In turn, this necessitates creating millions of noise-image pairs by running the full sampling steps of the teacher model, which is particularly costly for text-to-image synthesis. The regression loss also negates the key benefit of DMD's unpaired distribution matching objective, because it causes the student's quality to be upper-bounded by the teacher's.

In this paper, we show how to do away with DMD's regression loss, without compromising training stability. We then push the limits of distribution matching by integrating the GAN framework into DMD, and enable few-steps sampling with a novel training procedure, which we termed 'backward simulation'. Taken together, our contributions lead to state-of-the-art fast generative models that outperform their teacher, using as few as 4 sampling steps. Our method, which we call DMD2,

achieves state-of-the-art results in one-step image generation, setting a new benchmark with FID scores of 1.28 on ImageNet-64×64 and 8.35 on zero-shot COCO 2014. We demonstrate our approach's scalability by distilling from SDXL to produce high-quality megapixel images, establishing new standards among few-step methods.

In short, our contributions are as follows:

- We propose a new distribution matching distillation technique that does not require a regression loss for stable training, thereby eliminating the need for costly data collection, and allowing for more flexible and scalable training.

- We show that training instability in DMD [22] without regression loss stems from an insufficiently trained *fake diffusion critic*, and implement a two time-scale update rule to address this issue.

- We integrate a GAN objective into the DMD framework, where the discriminator is trained to distinguish samples from the student generator vs. *real* images. This additional supervision operates at the *distribution* level, which better aligns with DMD's distribution-matching philosophy than the original regression loss. It mitigates approximation errors in the teacher diffusion model and enhances image quality.

- While the original DMD only supports one-step students, we introduce a technique to support multi-step generators. Unlike previous multi-step distillation methods, we avoid the domain mismatch between training and inference by simulating inference-time generator inputs during training, thus improving overall performance.

## 2   Related Work

**Diffusion Distillation.** Recent diffusion acceleration techniques have focused on speeding up the generation process through distillation [9, 10, 13–20, 22, 23, 30]. They typically train a generator to approximate the ordinary differential equation (ODE) sampling trajectory of a teacher model, in fewer sampling steps. Notably, Luhman et al. [16] precompute a dataset of noise and images pairs, generated by the teacher using an ODE sampler, and use it to train the student to regress the mapping in a single network evaluation. Follow-up works like Progressive Distillation [10, 13] eliminate the need to precompute this paired dataset offline. They iteratively train a sequence of student models, each halving the number of sampling steps of its predecessor. A complementary technique, Instaflow [11] straightens the ODE trajectories, so they are easier to approximate with a one-step student. Consistency Distillation [9, 12, 19, 26, 31, 32], and TRACT [33], train student models so their outputs are self-consistent at any timesteps along the ODE trajectory, and thus consistent with the teacher.

**GANs.** Another line of research employs adversarial training to align the student with the teacher at a broader distribution level. In ADD [23], the generator, initialized with weights from a diffusion model, is trained using a projected GAN objective with an image-space classifier [34]. Building on this, LADD [24] utilizes a pre-trained diffusion model as the discriminator and operates in latent space, thus improving scalability and enabling higher-resolution synthesis. Inspired by DiffusionGAN [28, 29], UFOGen [25] introduces noise injection prior to the *real* vs. *fake* classification in the discriminator, to smooth out the distributions, which stabilizes the training dynamics. However, purely GAN-based methods often struggle to integrate classifier-free guidance directly. For instance, LADD uses diffusion-generated images with classifier-free guidance as real data in its GAN discriminator. Other approaches combine adversarial objectives with a distillation loss to preserve the original guided sampling trajectory. For instance, SDXL-Lightning [27] integrates a DiffusionGAN loss [25] with a progressive distillation objective [10, 13], while the Consistency Trajectory Model [26] combines a GAN [35] with an improved consistency distillation [9]. In contrast, our approach based on distribution matching [22, 36, 37] inherently integrates classifier-free guidance into the training supervision, significantly simplifying the training process.

**Score Distillation** was initially introduced in the context of text-to-3D synthesis [37–40], utilizing a pre-trained text-to-image diffusion model as a distribution matching loss. These methods optimize a 3D object by aligning rendered views with a text-conditioned image distribution, using the scores predicted by a pretrained diffusion model. Recent works have extended score distillation [37, 38, 41–43] to diffusion distillation [22, 30, 36, 44–46]. Notably, DMD [22] minimizes an approximate KL

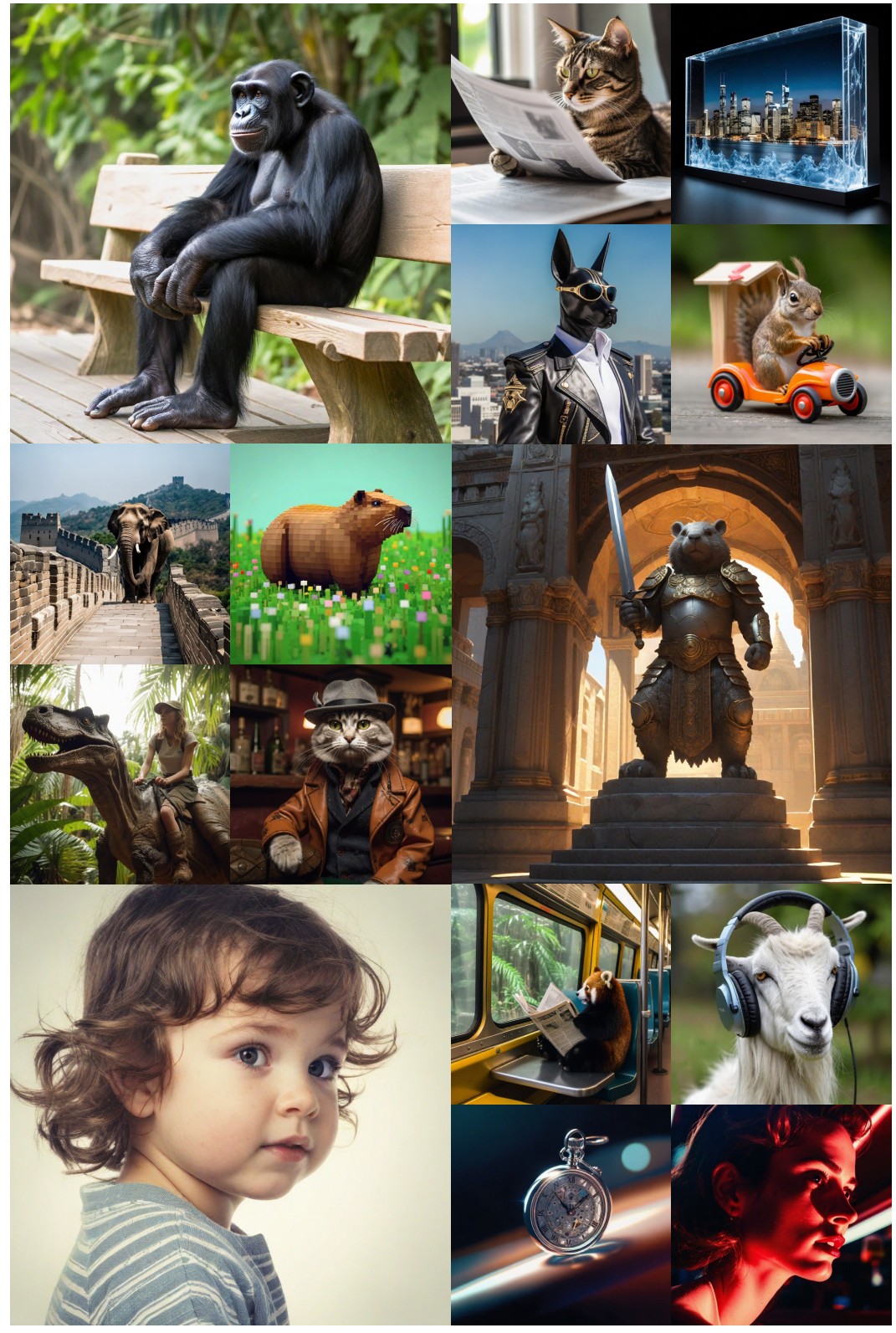

Figure 2: 1024×1024 samples produced by our 4-step generator distilled from SDXL. Please zoom in for details.

divergence, with its gradient represented as the difference between two score functions: one, fixed and pretrained, for the target distribution and another, trained dynamically, for the output distribution of the generator.

DMD parameterizes both score functions using diffusion models. This training objective proved more stable than GAN-based methods and has demonstrated superior performance in one-step image synthesis. An important caveat, DMD requires a regression loss for stability, calculated using precomputed noise-image pairs, similar to Luhman et al. [16]. Our work does away with this requirement. We introduce techniques to stabilize the DMD training procedure without the regression regularizer, thus significantly reducing the computational costs incurred by paired data precomputation. Furthermore, we extend DMD to support multi-step generation and integrate the strengths of both GANs and distribution matching approaches [22, 30, 36, 45], leading to state-of-the-art results in text-to-image synthesis.

## 3 Background: Diffusion and Distribution Matching Distillation

This section gives a brief overview of diffusion models and distribution matching distillation (DMD).

**Diffusion Models** generate images through iterative denoising. In the forward diffusion process, noise is progressively added to corrupt a sample $x \sim p_{\text{real}}$ from the data distribution into pure Gaussian noise over a predetermined number of steps $T$, so that, at each timestep $t$, the diffused samples follow the distribution $p_{\text{real},t}(x_t) = \int p_{\text{real}}(x)q(x_t|x)dx$, with $q_t(x_t|x) \sim \mathcal{N}(\alpha_t x, \sigma_t^2 \mathbf{I})$, where $\alpha_t, \sigma_t > 0$ are scalars determined by the noise schedule [47, 48]. The diffusion model learns to iteratively reverse the corruption process by predicting a denoised estimate $\mu(x_t, t)$, conditioned on the current noisy sample $x_t$ and the timestep $t$, ultimately leading to an image from the data distribution $p_{\text{real}}$. After training, the denoised estimate relates to the gradient of the data likelihood function, or score function [48] of the diffused distribution:

$$s_{\text{real}}(x_t, t) = \nabla_{x_t} \log p_{\text{real},t}(x_t) = -\frac{x_t - \alpha_t \mu_{\text{real}}(x_t, t)}{\sigma_t^2}. \tag{1}$$

Sampling an image typically requires dozens to hundreds of denoising steps [49–52].

**Distribution Matching Distillation (DMD)** distills a many-step diffusion models into a one-step generator $G$ [22] by minimizing the expectation over $t$ of approximate Kullback-Liebler (KL) divergences between the diffused target distribution $p_{\text{real},t}$ and the diffused generator output distribution $p_{\text{fake},t}$. Since DMD trains $G$ by gradient descent, it only requires the gradient of this loss, which can be computed as the difference of 2 score functions:

$$\nabla \mathcal{L}_{\text{DMD}} = \mathbb{E}_t \left( \nabla_\theta \text{KL}(p_{\text{fake},t} \| p_{\text{real},t}) \right) = -\mathbb{E}_t \left( \int \left( s_{\text{real}}(F(G_\theta(z), t), t) - s_{\text{fake}}(F(G_\theta(z), t), t) \right) \frac{dG_\theta(z)}{d\theta} \, dz \right), \tag{2}$$

where $z \sim \mathcal{N}(0, \mathbf{I})$ is a random Gaussian noise input, $\theta$ are the generator parameters, $F$ is the forward diffusion process (i.e., noise injection) with noise level corresponding to time step $t$, and $s_{\text{real}}$ and $s_{\text{fake}}$ are scores approximated using diffusion models $\mu_{\text{real}}$ and $\mu_{\text{fake}}$ trained on their respective distributions (Eq. (1)). DMD uses a frozen pre-trained diffusion model as $\mu_{\text{real}}$ (the teacher), and dynamically updates $\mu_{\text{fake}}$ while training $G$, using a denoising score-matching loss on samples from the one-step generator, i.e., fake data [22, 47].

Yin et al. [22] found that an additional regression term [16] was needed to regularize the distribution matching gradient (Eq. (2)) and achieve high-quality one-step models. For this, they collect a dataset of noise-image pairs $(z, y)$ where the image $y$ is generated using the teacher diffusion model, and a *deterministic* sampler [49, 50, 53], starting from the noise map $z$. Given the same input noise $z$, the regression loss compares the generator output with the teacher's prediction:

$$\mathcal{L}_{\text{reg}} = \mathbb{E}_{(z,y)} d(G_\theta(z), y), \tag{3}$$

where $d$ is a distance function, such as LPIPS [54] in their implementation. While gathering this data incurs negligible cost for small datasets like CIFAR-10, it becomes a significant bottleneck with large-scale text-to-image synthesis tasks, or models with complex conditioning [55–57]. For instance, generating one noise-image pair for SDXL [58] takes around 5 seconds, amounting to about 700 A100 days to cover the 12 million prompts in the LAION 6.0 dataset [59], as utilized by Yin et al. [22]. This dataset construction cost alone is already more than $4\times$ our total training compute (as

detailed in Appendix J). This regularization objective is also at odds with DMD's goal of matching the student and teacher in *distribution*, since it encourages adherence to the teacher's sampling paths.

# 4 Improved Distribution Matching Distillation

We revisit multiple design choices in the DMD algorithm [22] and identify significant improvements.

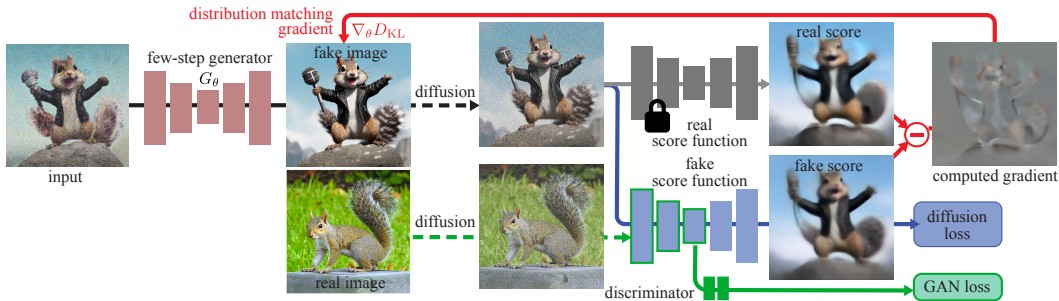

Figure 3: Our method distills a costly diffusion model (gray, right) into a one- or multi-step generator (red, left). Our training alternates between 2 steps: 1. optimizing the generator using the gradient of an implicit distribution matching objective (red arrow) and a GAN loss (green), and 2. training a score function (blue) to model the distribution of "fake" samples produced by the generator, as well as a GAN discriminator (green) to discriminate between fake samples and real images. The student generator can be a one-step or a multi-step model, as shown here, with an intermediate step input.

## 4.1 Removing the regression loss: true distribution matching and easier large-scale training

The regression loss [16] used in DMD [22] ensures mode coverage and training stability, but as we discussed in Section 3, it makes large-scale distillation cumbersome, and is at odds with the distribution matching idea, thus inherently limiting the performance of the distilled generator to that of the teacher model. Our first improvement is to remove this loss.

## 4.2 Stabilizing pure distribution matching with a Two Time-scale Update Rule

Naively omitting the regression objective, shown in Eq. (3), from DMD leads to training instabilities and significantly degrades quality (Tab. 3). For example, we observed that the average brightness, along with other statistics, of generated samples fluctuates significantly, without converging to a stable point (See Appendix G). We attribute this instability to approximation errors in the fake diffusion model $\mu_{\text{fake}}$, which does not track the fake score accurately, since it is dynamically optimized on the non-stationary output distribution of the generator. This causes approximation errors and biased generator gradients (as also discussed in [30]).

We address this using the two time-scale update rule inspired by Heusel et al. [60]. Specifically, we train $\mu_{\text{fake}}$ and the generator $G$ at different frequencies to ensure that $\mu_{\text{fake}}$ accurately tracks the generator's output distribution. We find that using 5 fake score updates per generator update, without the regression loss, provides good stability and matches the quality of the original DMD on ImageNet (Tab. 3) while achieving much faster convergence. Further analysis are included in Appendix G.

## 4.3 Surpassing the teacher model using a GAN loss and real data

Our model so far achieves comparable training stability and performance to DMD [22] without the need for costly dataset construction (Tab. 3). However, a performance gap remains between the distilled generator and the teacher diffusion model. We hypothesize this gap could be attributed to approximation errors in the real score function $\mu_{\text{real}}$ used in DMD, which would propagate to the generator and lead to suboptimal results. Since DMD's distilled model is never trained with real data, it cannot recover from these errors.

We address this issue by incorporating an additional GAN objective into our pipeline, where the discriminator is trained to distinguish between *real* images and images produced by our generator.

Trained using real data, the GAN classifier does not suffer from the teacher's limitation, potentially allowing our student generator to surpass it in sample quality. Our integration of a GAN classifier into DMD follows a minimalist design: we add a classification branch on top of the bottleneck of the fake diffusion denoiser (see Fig. 3). The classification branch and upstream encoder features in the UNet are trained by maximizing the standard non-saturating GAN objective:

$$\mathcal{L}_{\text{GAN}} = \mathbb{E}_{x \sim p_{\text{real}}, t \sim [0,T]}[\log D(F(x,t))] + \mathbb{E}_{z \sim p_{\text{noise}}, t \sim [0,T]}[-\log(D(F(G_\theta(z),t)))], \quad (4)$$

where $D$ is the discriminator, and $F$ is the forward diffusion process (i.e., noise injection) defined in Section 3, with noise level corresponding to time step $t$. The generator $G$ minimizes this objective. Our design is inspired by prior works that use diffusion models as discriminators [24, 25, 27]. We note that this GAN objective is more consistent with the distribution matching philosophy since it does not require paired data, and is independent of the teacher's sampling trajectories.

## 4.4  Multi-step generator

With the proposed improvements, we are able to match the performance of teacher diffusion models on ImageNet and COCO (see Tab. 1 and Tab. 5). However, we found that larger scale models like SDXL [58] remain challenging to distill into a one-step generator because of limited model capacity and a complex optimization landscape to learn the direct mapping from noise to highly diverse and detailed images. This motivated us to extend DMD to support multi-step sampling.

We fix a predetermined schedule with $N$ timestep $\{t_1, t_2, \ldots t_N\}$, identical during training and inference. During inference, at each step, we alternate between denoising and noise injection steps, following the consistency model [9], to improve sample quality. Specifically, starting from Gaussian noise $z_0 \sim \mathcal{N}(0, \mathbf{I})$, we alternate between denoising updates $\hat{x}_{t_i} = G_\theta(x_{t_i}, t_i)$, and forward diffusion steps $x_{t_{i+1}} = \alpha_{t_{i+1}} \hat{x}_{t_i} + \sigma_{t_{i+1}} \epsilon$ with $\epsilon \sim \mathcal{N}(0, \mathbf{I})$, until we obtain our final image $\hat{x}_{t_N}$. Our 4-step model uses the following schedule: 999, 749, 499, 249, for a teacher model trained with 1000 steps.

## 4.5  Multi-step generator simulation to avoid training/inference mismatch

Previous multi-step generators are typically trained to denoise noisy *real* images [23,24,27]. However, during inference, except for the first step, which starts from pure noise, the generator's input come from a previous generator sampling step $\hat{x}_{t_i}$. This creates a training-inference mismatch that adversely impacts quality (Fig. 4). We address this issue by replacing the noisy real images during training, with noisy synthetic images $x_{t_i}$ produced by the current student generator running several steps, similar to our inference pipeline (§ 4.4). This is tractable because, unlike the teacher diffusion model, our generator only runs for a few steps. Our generator then denoises these simulated images and the outputs are supervised with the proposed loss functions. Using noisy synthetic images avoids the mismatch and improves overall performance (See Sec. 5.3).

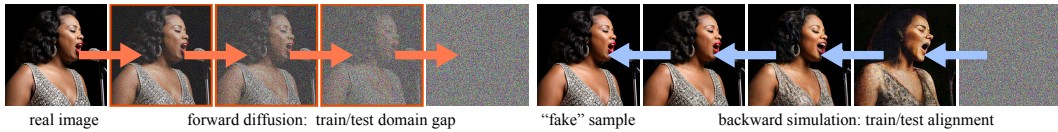

real image    forward diffusion:  train/test domain gap    "fake" sample    backward simulation: train/test alignment

Figure 4: Most multi-step distillation methods simulate intermediate steps using forward diffusion during training (left). This creates a mismatch with the inputs the model sees during inference. Our proposed solution (right) remedies the problem by simulating the inference-time backward process during training.

A concurrent work, Imagine Flash [61], proposed a similar technique. Their backward distillation algorithm shares our motivation of reducing the training and testing gap by using the student-generated images as the input to the subsequent sampling steps at training time. However, they do not entirely resolve the mismatch issue, because the teacher model of the regression loss now suffers from the training–test gap: it is never trained with synthetic images. This error is accumulated along the sampling path. In contrast, our distribution matching loss is not dependent on the input to the student model, alleviating this issue.

### 4.6 Putting everything together

In summary, our distillation method lifts DMD [22] stringent requirements for precomputed noise–image pairs. It further integrates the strength of GANs and supports multi-step generators. As shown in Fig. 3, starting from a pretrained diffusion model, we alternate between optimizing the generator $G_\theta$ to minimize the original distribution matching objective as well as a GAN objective, and optimizing the fake score estimator $\mu_{\text{fake}}$ using both a denoising score matching objective on the fake data, and the GAN classification loss. To ensure the fake score estimate is accurate and stable, despite being optimized on-line, we update it with higher frequency than the generator (5 steps vs. 1). A comparison of the training algorithms between DMD and DMD2 (Ours) is shown in Appendix Alg. 7.

## 5 Experiments

We evaluate our approach, DMD2, using several benchmarks, including class-conditional image generation on ImageNet-64×64 [62], and text-to-image synthesis on COCO 2014 [63] with various teacher models [1, 58]. We use the Fréchet Inception Distance (FID) [60] to measure image quality and diversity, and the CLIP Score [64] to evaluate text-to-image alignment. For SDXL models, we additionally report patch FID [27, 65], which measures FID on 299x center-cropped patches of each image, to assess high-resolution details. Finally, we conduct human evaluations to compare our approach with other state-of-the-art methods. Comprehensive evaluations confirm that distilled models trained using our approach outperform previous work, and even rival the performance of the teacher models. Detailed training and evaluation procedures are provided in the appendix.

### 5.1 Class-conditional Image Generation

Table 1 compares our model with recent baselines on ImageNet-64×64. With a single forward pass, our method significantly outperforms existing distillation techniques and even outperforms the teacher model using ODE sampler [53]. We attribute this remarkable performance to the removal of DMD's regression loss (Sec. 4.1 and 4.2), which eliminates the performance upper bound imposed by the ODE sampler, as well as our additional GAN term (Sec. 4.3), which mitigates the adverse impact of the teacher diffusion model's score approximation error.

Table 1: Image quality comparison on ImageNet-64×64.

| Method | # Fwd Pass ($\downarrow$) | FID ($\downarrow$) |
|---|---|---|
| BigGAN-deep [66] | 1 | 4.06 |
| ADM [67] | 250 | 2.07 |
| RIN [68] | 1000 | 1.23 |
| StyleGAN-XL [35] | 1 | 1.52 |
| Progress. Distill. [10] | 1 | 15.39 |
| DFNO [69] | 1 | 7.83 |
| BOOT [20] | 1 | 16.30 |
| TRACT [33] | 1 | 7.43 |
| Meng et al. [13] | 1 | 7.54 |
| Diff-Instruct [36] | 1 | 5.57 |
| Consistency Model [9] | 1 | 6.20 |
| iCT-deep [12] | 1 | 3.25 |
| CTM [26] | 1 | 1.92 |
| DMD [22] | 1 | 2.62 |
| **DMD2 (Ours)** | 1 | **1.51** |
| **+longer training (Ours)** | 1 | **1.28** |
| EDM (Teacher, ODE) [53] | 511 | 2.22 |
| EDM (Teacher, SDE) [53] | 511 | 1.36 |

Table 2: Image quality comparison with SDXL backbone on 10K prompts from COCO 2014.

| Method | # Fwd Pass ($\downarrow$) | FID ($\downarrow$) | Patch FID ($\downarrow$) | CLIP ($\uparrow$) |
|---|---|---|---|---|
| LCM-SDXL [32] | 1 | 81.62 | 154.40 | 0.275 |
|  | 4 | 22.16 | 33.92 | 0.317 |
| SDXL-Turbo [23] | 1 | 24.57 | 23.94 | **0.337** |
|  | 4 | 23.19 | 23.27 | 0.334 |
| SDXL Lightning [27] | 1 | 23.92 | 31.65 | 0.316 |
|  | 4 | 24.46 | 24.56 | 0.323 |
| **DMD2 (Ours)** | 1 | **19.01** | 26.98 | 0.336 |
|  | 4 | 19.32 | **20.86** | 0.332 |
| SDXL Teacher, cfg=6 [58] | 100 | 19.36 | 21.38 | 0.332 |
| SDXL Teacher, cfg=8 [58] | 100 | 20.39 | 23.21 | 0.335 |

## 5.2 Text-to-Image Synthesis

We evaluate DMD2's text-to-image generation performance on zero-shot COCO 2014 [63]. Our generators are trained by distilling SDXL [58] and SD v1.5 [1], respectively, using a subset of 3 million prompts from LAION-Aesthetics [59]. Additionally, we collect 500k images from LAION-Aesthetic as training data for the GAN discriminator. Table 2 summarizes distillation results for the SDXL model. Our 4-step generator produces high quality and diverse samples, achieving a FID score of 19.32 and a CLIP score of 0.332, rivaling the teacher diffusion model for both image quality and prompt coherence. To further verify our method's effectiveness, we conduct an extensive user study comparing our model's output with those from the teacher model and existing distillation methods. We use a subset of 128 prompts from PartiPrompts [70] following LADD [24]. For each comparison, we ask a random set of five evaluators to choose the image that is more visually appealing, as well as the one that better represents the text prompt. Details about the human evaluation are included in Appendix L. As shown in Figure 5, our model achieves much higher user preferences than baseline approaches. Notably, our model outperforms its teacher in image quality for 24% of samples and achieves comparable prompt alignment, while requiring $25\times$ fewer forward passes (4 vs 100). Qualitative comparisons are shown in Figure 6. Results for SDv1.5 are provided in Table 5 in Appendix D. Similarly, one-step model trained using DMD2 outperforms all previous diffusion acceleration approaches, achieving a FID score of 8.35, representing a significant 3.14-point improvement over the original DMD method [22]. Our results also surpass the teacher models that uses a 50-step PNDM sampler [50].

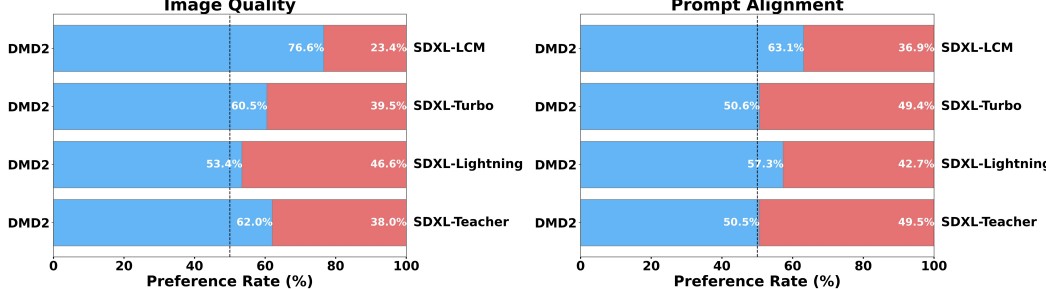

Figure 5: User study comparing our distilled model with its teacher and competing distillation baselines [23, 27, 31]. All distilled models use 4 sampling steps, the teacher uses 50. Our model achieves the best performance for both image quality and prompt alignment.

## 5.3 Ablation Studies

Table 3: Ablation studies on ImageNet. TTUR stands for two-timescale update rule.

| DMD | No Regress. | TTUR | GAN | FID (↓) |
|-----|-----|-----|-----|-----|
| ✓ | | | | 2.62 |
| ✓ | ✓ | | | 3.48 |
| ✓ | ✓ | ✓ | | 2.61 |
| ✓ | ✓ | ✓ | ✓ | **1.51** |
| | | | ✓ | 2.56 |
| | | ✓ | ✓ | 2.52 |

Table 4: Ablation studies with SDXL backbone on 10K prompts from COCO 2014.

| Method | FID (↓) | Patch FID (↓) | CLIP (↑) |
|-----|-----|-----|-----|
| w/o GAN | 26.90 | 27.66 | 0.328 |
| w/o Distribution Matching | **13.77** | 27.96 | 0.307 |
| w/o Backward Simulation | 20.66 | 24.21 | 0.332 |
| **DMD2 (Ours)** | 19.32 | **20.86** | **0.332** |

Table 3 ablates different components of our proposed method on ImageNet. Simply removing the ODE regression loss from the original DMD results in a degraded FID of 3.48 due to training instability (see further analysis in Appendix G). However, incorporating our Two Time-scale Update Rule (TTUR, Sec. 4.2) mitigates this performance drop, matching the DMD baseline performance without requiring additional dataset construction. Adding our GAN loss achieves a further 1.1-point improvement in FID. Our integrated approach surpasses the performance of using GAN alone (without distribution matching objective), and adding the two-timescale update rule to GAN alone does not improve it, highlighting the effectiveness of combining distribution matching with GANs in a unified framework.

**A photo of llama wearing sunglasses standing on the deck of a spaceship with the Earth in the background.**

**a shiba inu wearing a beret and black turtleneck**

**a young girl playing piano**

**A train ride in the monsoon rain in Kerala. With a Koala bear wearing a hat looking out of the window. There is a lot of coconut trees out of the window.**

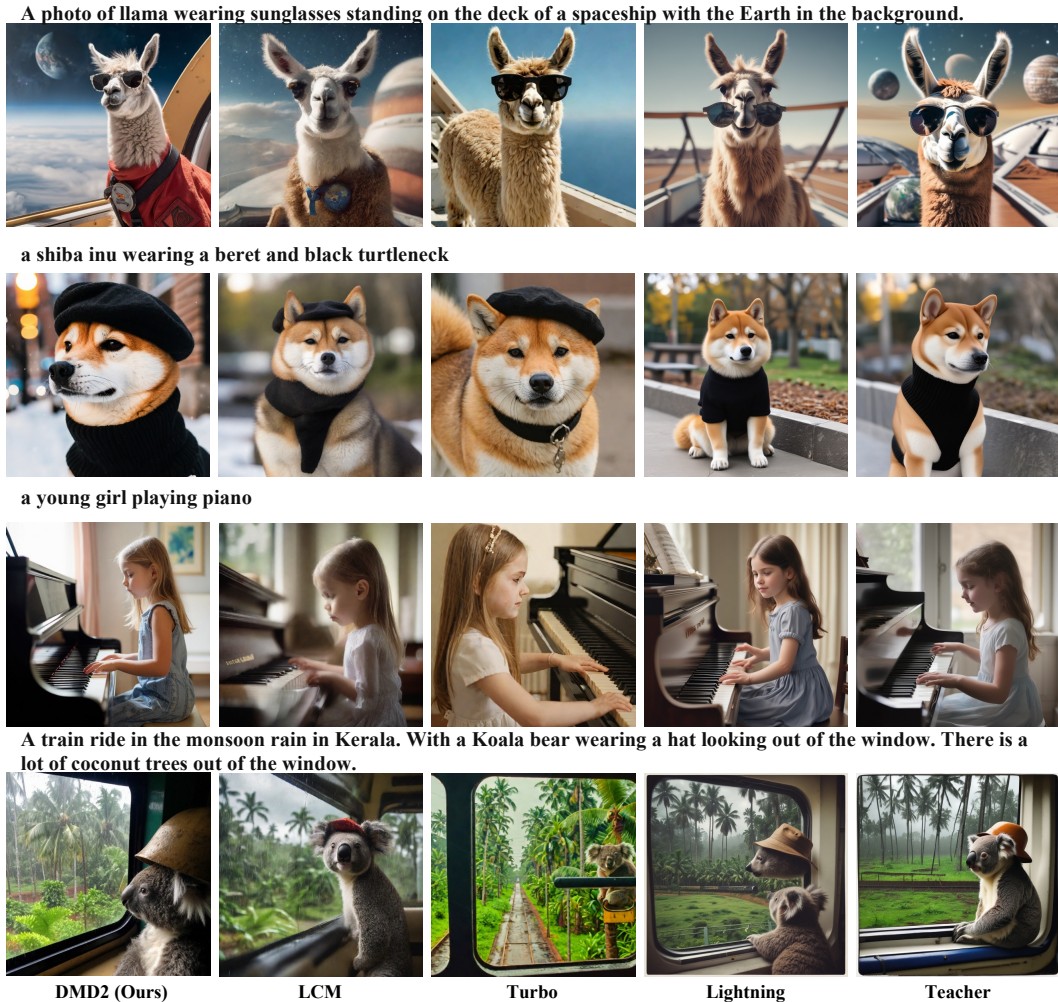

| DMD2 (Ours) | LCM | Turbo | Lightning | Teacher |

Figure 6: Visual comparison between our model, the SDXL teacher, and selected competing methods [23, 27, 31]. All distilled models use 4 sampling steps while the teacher model uses 50 sampling steps with classifier-free guidance. All images are generated using identical noise and text prompts. Our model produces images with superior realism and text alignment. (Zoom in for details.) More comparisons are available in Appendix Figure 11.

In Table 4, we ablate the influence of the GAN term (Sec. 4.3), distribution matching objective (Eq. 2), and backward simulation (Sec. 4.4) for distilling the SDXL model into a four-step generator. Qualitative results are shown in Appendix Figure. 8. In the absence of the GAN loss, our baseline model produces oversaturated and oversmoothed images (Appendix Fig. 8 third column). Similarly, eliminating distribution matching objective (Eq. 2) reduces our approach to a pure GAN-based method, which struggles with training stability [71, 72]. Moreover, pure GAN-based methods also lack a natural way to incorporate classifier-free guidance [73], essential for high-quality text-to-image synthesis [1, 2]. Consequently, while GAN-based methods achieve the lowest FID by closely matching the real distribution, they significantly underperform in text alignment and aesthetic quality (Appendix Fig. 8 second column). Likewise, omitting the backward simulation leads to worse image quality, as indicated by the degraded patch FID score.

# 6 Acknowledgements

We extend our gratitude to Minguk Kang and Seungwook Kim for their assistance in setting up the human evaluation. We also thank Zeqiang Lai for suggesting the timestep shift technique used in our one-step generator. Additionally, we are grateful to our friends and colleagues for their insightful discussions and valuable comments. This work was supported by the National Science Foundation under Cooperative Agreement PHY-2019786 (The NSF AI Institute for Artificial Intelligence and Fundamental Interactions, http://iaifi.org/), by NSF Grant 2105819, by NSF CISE award 1955864, and by funding from Google, GIST, Amazon, and Quanta Computer.

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

# Table of Contents

## A  Limitations

While achieving superior image quality and text alignment, our distilled generator experiences a slight degradation in image diversity compared to the teacher models (see Appendix F). Additionally, our generator still requires four steps to match the quality of the largest SDXL model. These limitations, while not unique to our model, highlight areas for further improvement. Like most previous distillation methods, we use a fixed guidance scale during training, limiting user flexibility. Introducing a variable guidance scale [13, 31] could be a promising direction for future research. Furthermore, our methods are optimized for distribution matching; incorporating human feedback or other reward functions could further enhance performance [17, 74]. Lastly, training large-scale generative models is computationally intensive, making it inaccessible for most researchers. We hope our efficient approach and optimized, user-friendly codebase will help democratize future research in this field.

## B  Code, dataset, and more results

We are continually updating the paper and related materials. For access to our code, model, and the latest version of the paper with expanded results and analysis, please visit our project website.

Table 5: Sample quality comparison on 30K prompts from COCO 2014.

| Family | Method | Resolution (↑) | Latency (↓) | FID (↓) |
|---|---|---|---|---|
| **Original, unaccelerated** | DALL·E [78] | 256 | - | 27.5 |
| | DALL·E 2 [3] | 256 | - | 10.39 |
| | Parti-750M [70] | 256 | - | 10.71 |
| | Parti-3B [70] | 256 | 6.4s | 8.10 |
| | Make-A-Scene [79] | 256 | 25.0s | 11.84 |
| | GLIDE [80] | 256 | 15.0s | 12.24 |
| | LDM [1] | 256 | 3.7s | 12.63 |
| | Imagen [4] | 256 | 9.1s | 7.27 |
| | eDiff-I [5] | 256 | 32.0s | **6.95** |
| **GANs** | LAFITE [81] | 256 | 0.02s | 26.94 |
| | StyleGAN-T [82] | 512 | 0.10s | 13.90 |
| | GigaGAN [72] | 512 | 0.13s | **9.09** |
| **Accelerated diffusion** | DPM++ (4 step) [51] | 512 | 0.26s | 22.36 |
| | UniPC (4 step) [83] | 512 | 0.26s | 19.57 |
| | LCM-LoRA (4 step) [32] | 512 | 0.19s | 23.62 |
| | InstaFlow-0.9B [11] | 512 | 0.09s | 13.10 |
| | SwiftBrush [45] | 512 | 0.09s | 16.67 |
| | HiPA [84] | 512 | 0.09s | 13.91 |
| | UFOGen [25] | 512 | 0.09s | 12.78 |
| | SLAM (4 step) [18] | 512 | 0.19s | 10.06 |
| | DMD [22] | 512 | 0.09s | 11.49 |
| | **DMD2 (Ours)** | 512 | 0.09s | **8.35** |
| **Teacher** | SDv1.5 (50 step, cfg=3, ODE) [1, 50] | 512 | 2.59s | 8.59 |
| | SDv1.5 (200 step, cfg=2, SDE) [1, 47] | 512 | 10.25s | 7.21 |

## C  Broader Impact

Our work on improving the efficiency and quality of diffusion model has several potential societal impacts, both positive and negative. On the positive side, the advancements in fast image synthesis can significantly benefit various creative industries. These models can enhance graphic design, animation, and digital art by providing artists with powerful tools to generate high-quality visuals efficiently. Additionally, improved text-to-image synthesis capabilities can be used in education and entertainment, enabling the creation of personalized learning materials and immersive experiences.

However, potential negative societal impacts must be considered. Misuse risks include generating misinformation and creating fake profiles, which could spread false information and manipulate public opinion. Deploying these technologies could result in biases that unfairly impact specific groups, especially if models are trained on biased datasets, potentially perpetuating or amplifying existing societal biases. To mitigate these risks, we are interested in developing monitoring mechanisms to detect and prevent misuse [75, 76] and methods to enhance output diversity and fairness [77].

## D  SD v1.5 Results

Table 5 presents detailed comparisons between our one-step generator distilled from SD v1.5 and competing approaches.

## E  Comparison with DMD1

A comparison between DMD and DMD2 is shown in Algorithm 7.

## F  Text-to-Image Synthesis Further Analysis

Qualitative ablation results using SDXL backbone are shown in Figure 8. Additionally, we compare the image diversity of our 4-step generator with other competing approaches distilled from SDXL [23,

Table 6: Image quality and diversity comparison with SDXL backbone.

| Method | # Fwd Pass ($\downarrow$) | FID ($\downarrow$) | Patch FID ($\downarrow$) | CLIP ($\uparrow$) | Diversity Score ($\uparrow$) |
|---|---|---|---|---|---|
| LCM-SDXL [32] | 4 | 22.16 | 33.92 | 0.317 | 0.61 |
| SDXL-Turbo [23] | 4 | 23.19 | 23.27 | **0.334** | 0.58 |
| SDXL-Lightning [27] | 4 | 24.46 | 24.56 | 0.323 | **0.63** |
| **DMD2 (Ours)** | 4 | **19.32** | **20.86** | 0.332 | 0.61 |
| SDXL-Teacher, cfg=6 [58] | 100 | 19.36 | 21.38 | 0.332 | 0.64 |
| SDXL-Teacher, cfg=8 [58] | 100 | 20.39 | 23.21 | 0.335 | 0.64 |

| Method | ImageReward | Aesthetic Score |
|---|---|---|
| SDXL | 0.86 | 6.16 |
| **DMD2** | **1.07** | **6.30** |

Table 7: Comparison of ImageReward and Aesthetic Score for Different Methods

27, 31]. We employ an LPIPS-based diversity score, similar to that used in multi-modal image-to-image translation [85, 86]. Specifically, we generate four images per prompt and calculate the average pairwise LPIPS distance [54]. For this evaluation, we use the LADD [24] subset of PartiPrompts [70]. We also report the FID and CLIP score measured on 10K prompts from COCO 2014 on the side. Table 6 summarizes the results. Table 7 provides further comparisons using image reward [87] and aesthetic score metrics [59]. Our model achieves the best image quality, indicated by the lowest FID and Patch FID scores. We also achieve text alignment comparable to SDXL-Turbo while attaining a better diversity score. While SDXL-Lightning [27] exhibits a higher diversity score than our approach, it suffers from considerably worse text alignment, as reflected by the lower CLIP score and human evaluation (Fig. 5). This suggests that the improved diversity is partially due to random outputs lacking prompt coherence. We note that it is possible to increase the diversity of our model by raising the weights for the GAN objective, which aligns with the more diverse unguided distribution. Further investigation into finding the optimal balance between distribution matching and the GAN objective is left for future work.

## G    Two Time-scale Update Rule Further Analysis

In Section 4.2, we discuss that updating the fake score multiple times (5 updates) per generator update leads to better stability. Here, we provide further analysis. Figure 9 visualizes pixel brightness variations throughout training. The baseline approach, which omits the regression objective from DMD and uses just 1 fake score update, results in significant training instability, as evidenced by periodic fluctuations in pixel brightness. In contrast, our two time-scale update rule with 5 fake score updates per generator update stabilizes the training and leads to better sample quality, as shown in Tab. 3.

We further examine the influence of the update frequency for the fake diffusion model $\mu_{\text{fake}}$ in Figure 10. An update frequency of 1 fake diffusion update per generator update corresponds to the naive baseline (red line) and suffers from training instability. Although a frequency of 10 updates (magenta line) provides excellent stability, it significantly slows down the training process. We found that a moderate frequency of 5 updates (green line) achieves the best balance between stability and convergence speed on ImageNet. Our approach proves more effective than using asynchronous learning rates [60] (cyan line) and converges significantly faster than the original DMD method that employs a regression loss [22] (dark blue line). For new models and datasets, we recommend adjusting the iteration number to the smallest value that ensures the stability of general image statistics, such as pixel brightness.

## H    Additional Text-to-Image Synthesis Results

Additional visual comparisons for the 4-step distilled models are shown in Figure 11. Sample outputs from our one-step generator are presented in Figure 12.

**Algorithm 1: DMD** (original)

**Input:** Pretrained real diffusion model $\mu_{\text{real}}$, paired ODE solution pairs $\mathcal{D} = \{z_{\text{ref}}, y_{\text{ref}}\}$
**Output:** Trained generator $G$

```
1  // Initialize generator & fake score model
2  G ← copyWeights(μ_real)
3  μ_fake ← copyWeights(μ_real)
4  while train do
5  |   // Generate batch of images
6  |   Sample z ~ N(0, I)^B;  (z_ref, y_ref) ~ D;
7  |   x ← G(z); x_ref ← G(z_ref);
8  |   // 1. Update generator
9  |   L_KL ← distributionMatchingLoss(μ_real, μ_fake, x);
10 |   L_reg ← LPIPS(x_ref, y_ref);
11 |   L_G ← L_KL + λ_reg L_reg;
12 |   G ← update(G, L_G);
13 |   // 2. Update fake score model
14 |   Sample t ~ U(0, 1);
15 |   x_t ← forwardDiffusion(stopgrad(x), t);
16 |   L_denoise ← denoisingLoss(μ_fake(x_t, t), stopgrad(x));
17 |   μ_fake ← update(μ_fake, L_denoise);
18 end while
```

**Algorithm 2: DMD2** (ours)

**Input:** Pretrained real diffusion model $\mu_{\text{real}}$, real image dataset $\mathcal{D}_{\text{real}}$,
`update_G_freq`: frequency of generator updates
**Output:** Trained generator $G$

```
1  // Initialize generator, fake score model,
      discriminator
2  G ← copyWeights(μ_real)
3  μ_fake ← copyWeights(μ_real)
4  D ← initializeDiscriminator()
5  for iteration = 1 to maxIters do // main training loop
6  |   // (a) Generate batch of images
7  |   Sample z ~ N(0, I)^B;
8  |   if multi-step then
9  |   |   x ← G(multiStepSampling(G, z));
          // simulate multi-step inference
10 |   end if
11 |   else
12 |   |   x ← G(z)
13 |   end if
14 |   // (b) Update G only once per
          update_G_freq iterations
15 |   if iteration % update_G_freq == 0 then
16 |   |   // (b1) Distribution matching loss
17 |   |   L_KL ← distributionMatchingLoss(μ_real, μ_fake, x);
18 |   |   // (b2) GAN loss term
19 |   |   L_GAN ← -E[log D(F(G(z), t))];
20 |   |   // (b3) Final generator loss
21 |   |   L_G ← L_KL + λ_GAN L_GAN;
22 |   |   // (b4) Apply gradient update to G
23 |   |   G ← update(G, L_G);
24 |   end if
25 |   Sample real images x_real ~ D_real;
26 |   Sample t ~ U(0, 1);
27 |   // (c) Update μ_fake (fake score) via
          denoising on fake data
28 |   x_t ← forwardDiffusion(stopgrad(x), t);
29 |   L_denoise ← denoisingLoss(μ_fake(x_t, t), stopgrad(x));
30 |   μ_fake ← update(μ_fake, L_denoise);
31 |   // (d) Update discriminator D (GAN
          classification)
32 |   L_GAN-D ← E[log D(F(x_real, t))] + E[log(1 -
          D(F(stopgrad(x), t)))];
33 |   D ← update(D, L_GAN-D);
34 end for
```

Figure 7: Side-by-side comparison of **DMD** (left) and our improved **DMD2** (right).

# I  ImageNet Visual Results

In Figure 13, we present qualitative results obtained from our one-step distilled model trained on the ImageNet dataset.

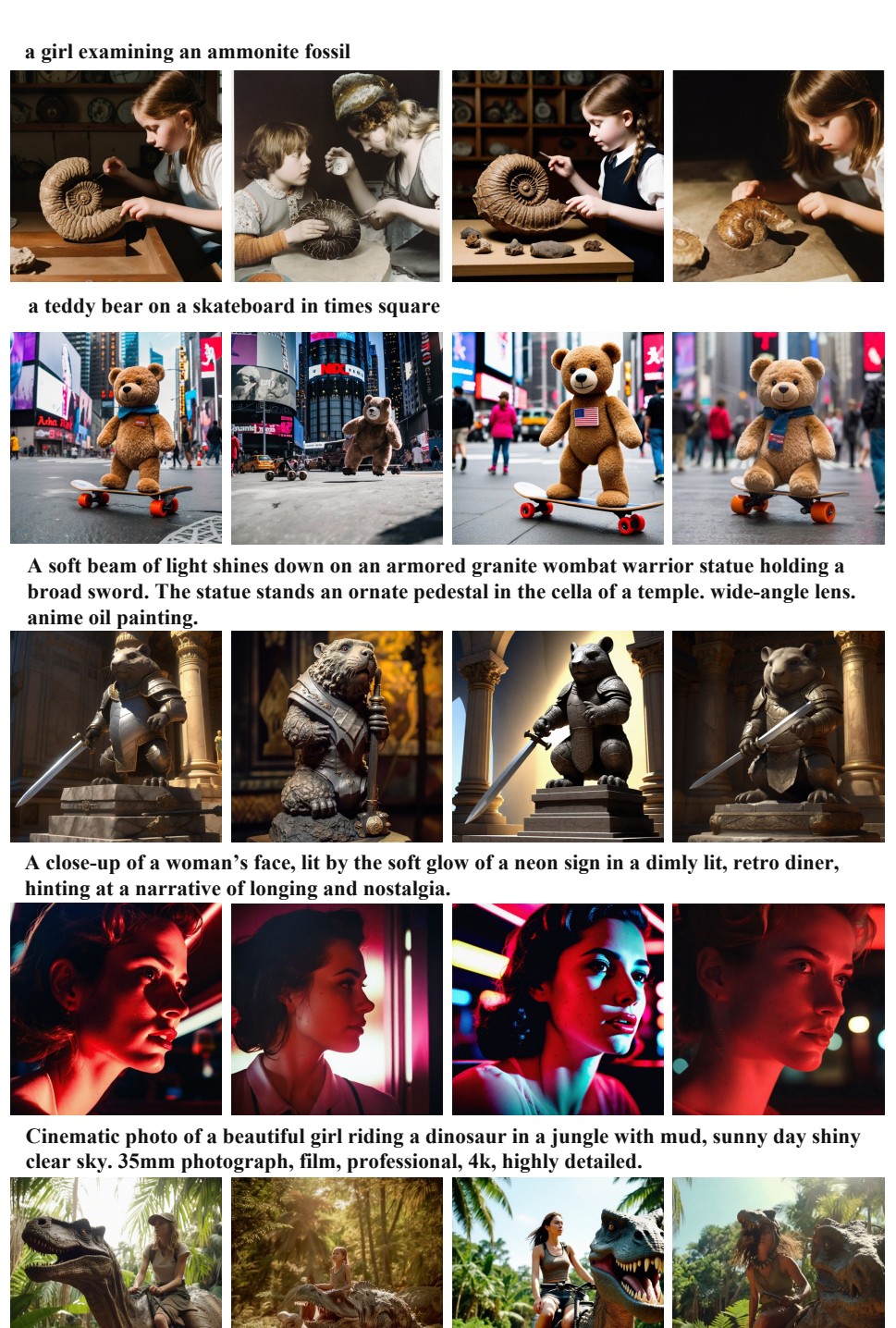

**a girl examining an ammonite fossil**

**a teddy bear on a skateboard in times square**

**A soft beam of light shines down on an armored granite wombat warrior statue holding a broad sword. The statue stands an ornate pedestal in the cella of a temple. wide-angle lens. anime oil painting.**

**A close-up of a woman's face, lit by the soft glow of a neon sign in a dimly lit, retro diner, hinting at a narrative of longing and nostalgia.**

**Cinematic photo of a beautiful girl riding a dinosaur in a jungle with mud, sunny day shiny clear sky. 35mm photograph, film, professional, 4k, highly detailed.**

| **DMD2 (Ours)** | **w/o Distribution Matching** | **w/o GAN** | **w/o Backward Simulation** |

Figure 8: SDXL Qualitative Ablations. All images are generated using identical noise and text prompts. Removing the distribution matching objective significantly degrades aesthetic quality and text alignment. Omitting the GAN loss results in oversaturated and overly smoothed images. The baseline without backward simulation produces images of lower quality.

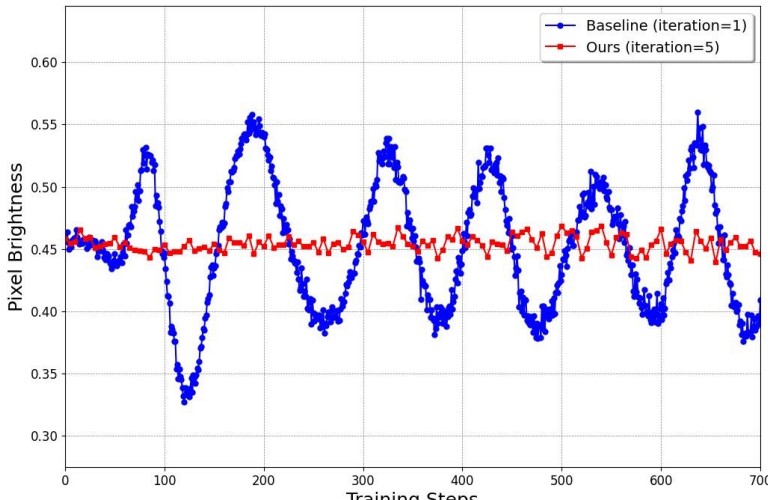

Figure 9: Visualization of pixel brightness variations throughout training. The baseline approach, which naively removes the regression loss from the original DMD [22], suffers from significant training instability, leading to fluctuating general image statistics like the overall pixel brightness. In contrast, our two time-scale update rule, which optimizes the fake diffusion model five times per generator update, significantly stabilizes training and enhances sample quality.

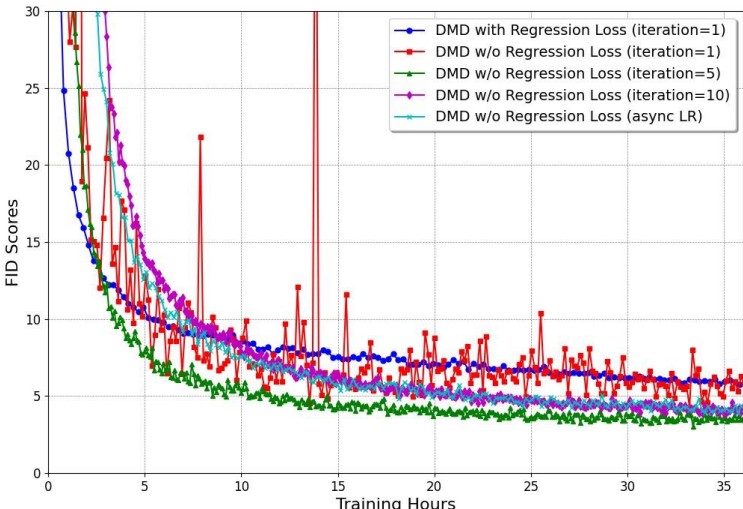

Figure 10: Visualization of FID score progression during training. Naively removing the regression loss leads to training instability (red line). A two time-scale update rule with five fake diffusion critic updates per generator update stabilizes training and is more effective than using a larger number of fake diffusion updates or an asynchronous learning rate where the fake diffusion model uses a learning rate 5 times larger than the generator. The model trained with our two time-scale update rule (green) also converges significantly faster than the original DMD method with a regression loss (dark blue), even though TTUR performs less number of the generator weight updates.

**an orange wearing a cowboy hat**

**A bald eagle made of chocolate powder, mango, and whipped cream**

**a pumpkin on a man's head**

**A punk rock squirrel in a studded leather jacket shouting into a microphone while standing on a boulder**

**a cat reading a newspaper**

| DMD2 (Ours) | LCM | Turbo | Lightning | Teacher |

Figure 11: Additional visual comparison between our model, the SDXL teacher, and selected competing methods [23, 27, 31]. All distilled models use 4 sampling steps while the teacher model uses 50 sampling steps with classifier-free guidance. All images are generated using identical noise and text prompts. Our model produces images with superior realism and text alignment. Please zoom in for details.

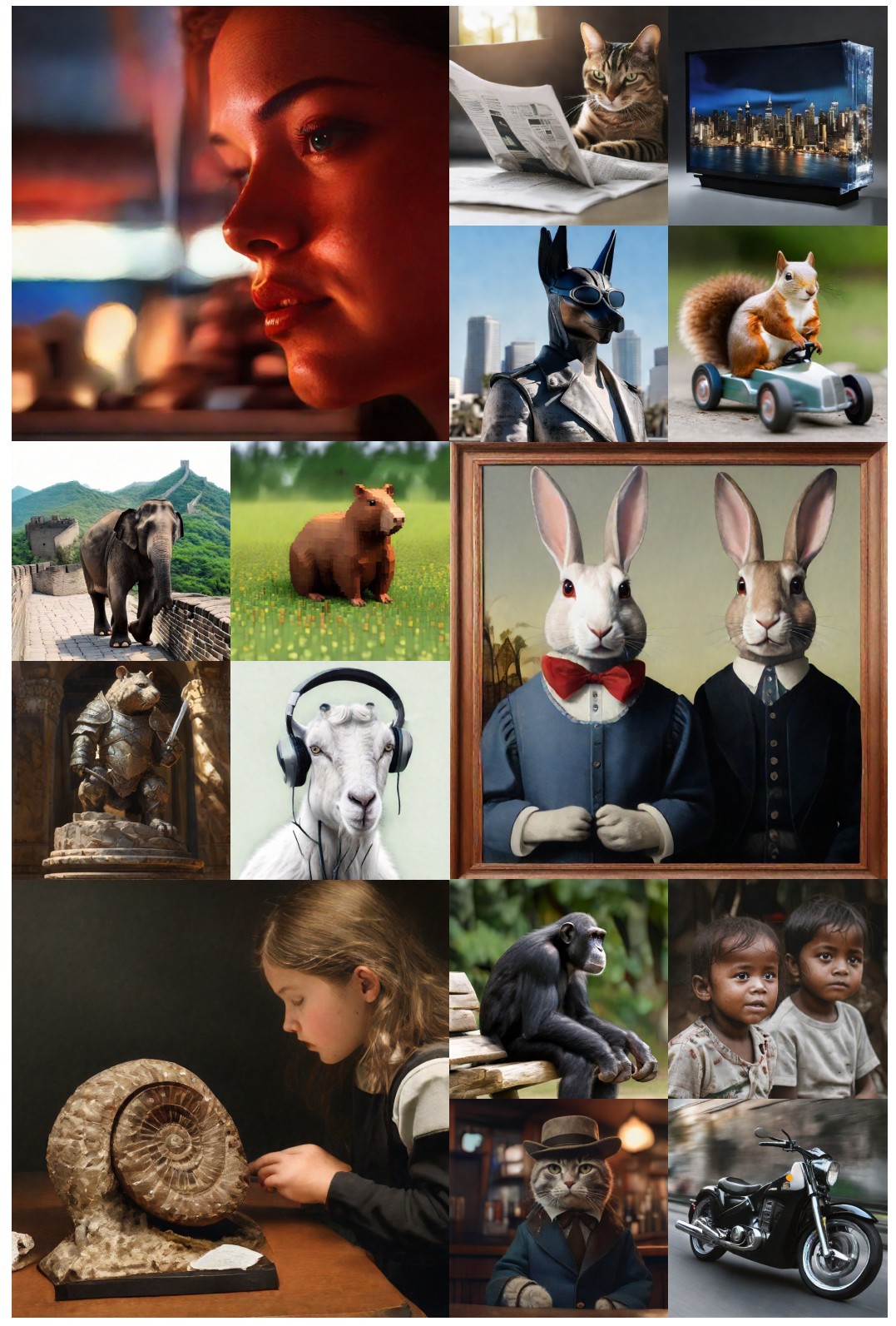

Figure 12: Additional 1024×1024 samples produced by our 1-step generator distilled from SDXL. Please zoom in for details.

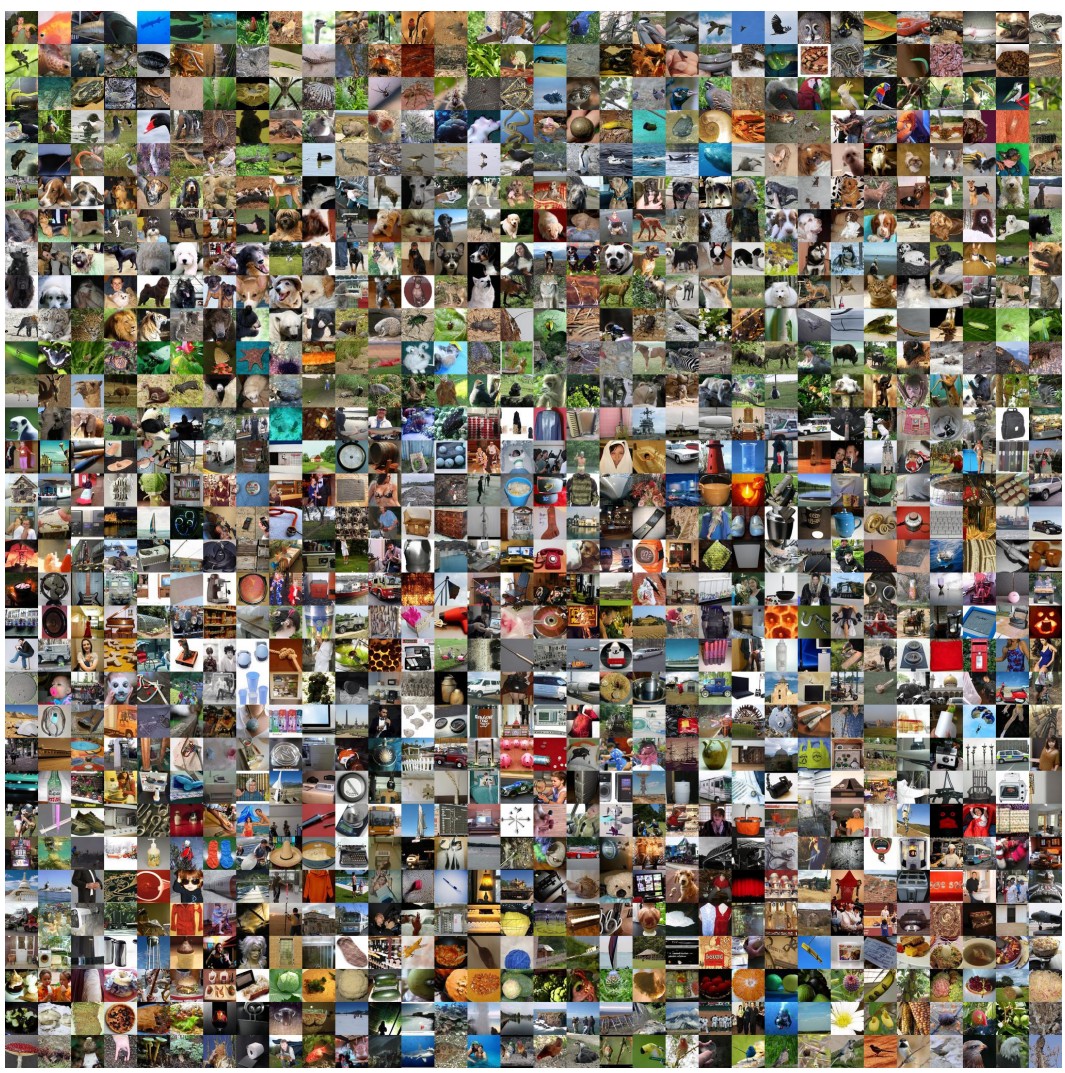

Figure 13: One-step samples from our generator trained on ImageNet (FID=1.28). Please zoom in for details.

# J  Implementation Details

This section outlines key aspects of the implementation. All results are reproducible using our open-source training and evaluation code. Generally, we employ the maximum batch size supported by our compute resources. The learning rate is set to the highest stable value, ensuring no divergent loss within the initial 500 iterations. We determine the TTUR iteration count based on the minimum needed for stability, while the guidance scale is selected to optimize performance for the teacher model.

## J.1  GAN Classifier Design

Our GAN classifier design is inspired by SDXL-Lightning [27]. Specifically, we attach a prediction head to the middle block output of the fake diffusion model. The prediction head consists of a stack of $4 \times 4$ convolutions with a stride of 2, group normalization, and SiLU activations. All feature maps are downsampled to $4 \times 4$ resolution, followed by a single convolutional layer with a kernel size and stride of 4. This layer pools the feature maps into a single vector, which is then passed to a linear projection layer to predict the classification result.

## J.2  ImageNet

Our ImageNet implementation closely follows the DMD paper [22]. Specifically, we distill a one-step generator from the EDM pretrained model [53], released under the CC BY-NC-SA 4.0 License. For the standard training setup, we use the AdamW optimizer [88] with a learning rate of $2 \times 10^{-6}$, a weight decay of 0.01, and beta parameters (0.9, 0.999). We use a batch size of 280 and train the model on 7 A100 GPUs for 200K iterations, which takes approximately 2 days. The number of fake diffusion model update per generator update is set to 5. The weight for the GAN loss is set to $3 \times 10^{-3}$. For the extended training setup shown in Table 1, we first pretrain the model without GAN loss for 400K iterations. We then resume from the best checkpoint (as measured by FID), enable the GAN loss with a weight of $3 \times 10^{-3}$, reduce the learning rate to $5 \times 10^{-7}$, and continue training for an additional 150K iterations. The total training time for this run is approximately 5 days.

## J.3  SD v1.5

We distill a one-step generator from the SD v1.5 model [1], released under the CreativeML Open RAIL-M license, using prompts from the LAION-Aesthetic 6.25+ dataset [59]. Additionally, we collect 500K images from LAION-Aesthetic 5.5+ as training data for the GAN discriminator, filtering out images smaller than $1024 \times 1024$ and those containing unsafe content. Our training process involves two stages. In the first stage, we disable the GAN loss and use the AdamW optimizer with a learning rate of $1 \times 10^{-5}$, a weight decay of 0.01, and beta parameters of (0.9, 0.999). The fake diffusion model is updated 10 times per generator update. We set the guidance scale for the real diffusion model to be 1.75. We use a batch size of 2048 and train the model on 64 A100 GPUs for 40K iterations. In the second stage, we enable the GAN loss with a weight of $10^{-3}$, reduce the learning rate to $5 \times 10^{-7}$, and continue training for an additional 5K iterations. The total training time is approximately 26 hours.

## J.4  SDXL

We train both one-step and four-step generators by distilling from the SDXL model [58], released under the CreativeML Open RAIL++-M License. For the one-step generator, we observed similar block noise artifacts as reported in SDXL-Lightning [27] and Pixart-Sigma [89]. We addressed this by adopting the timestep shift technique from OpenDMD [90] and Pixart-Sigma [89], setting the conditioning timestep to 399. Additionally, we initialized the one-step generator by pretraining it with a regression loss using a small set of 10K pairs for a short period. These adjustments are not necessary for the multi-step model or other backbones, suggesting this issue might be specific to SDXL. Similar to SD v1.5, we use prompts from the LAION-Aesthetic 6.25+ dataset [59] and collect 500K images from LAION-Aesthetic 5.5+ as training data for the GAN discriminator, filtering out images smaller than $1024 \times 1024$ and those containing unsafe content. The generator is trained using the AdamW optimizer with a learning rate of $5 \times 10^{-7}$, a weight decay of 0.01, and beta parameters of (0.9, 0.999). The fake diffusion model is updated 5 times per generator update. We set the guidance

scale for the real diffusion model to be 8. We use a batch size of 128 and train the model on 64 A100 GPUs for 20K iterations for the 4-step generator and 25K iterations for the 1-step generator, taking approximately 60 hours.

## K  Evaluation Details

For the COCO experiments, we follow the exact evaluation setup as GigaGAN [72] and DMD [22]. For the results presented in Table 5, we use 30K prompts from the COCO 2014 validation set and generate the corresponding images. The outputs are downsampled to 256×256 and compared with 40,504 real images from the same validation set using clean-FID [91]. For the results presented in Table 2, we use a random set of 10K prompts from the COCO 2014 validation set and generate the corresponding images. The outputs are downsampled to 512×512 and compared with the corresponding 10K real images from the validation set with the same prompts. We compute the CLIP score using the OpenCLIP-G backbone. For the ImageNet results, we generate 50,000 images and calculate the FID statistics using EDM's evaluation code [53].

## L  User Study Details

To conduct the human preference study, we use the Prolific platform (https://www.prolific.com). We use 128 prompts from the LADD [24] subset of PartiPrompts [70]. All approaches generate corresponding images, which are presented in pairs to human evaluators to measure aesthetic and prompt alignment preference. The specific questions and interface are shown in Figure 14. Consent is obtained from the voluntary participants. We manually verify that all generated images contain standard visual content that poses no risks to the study participants.

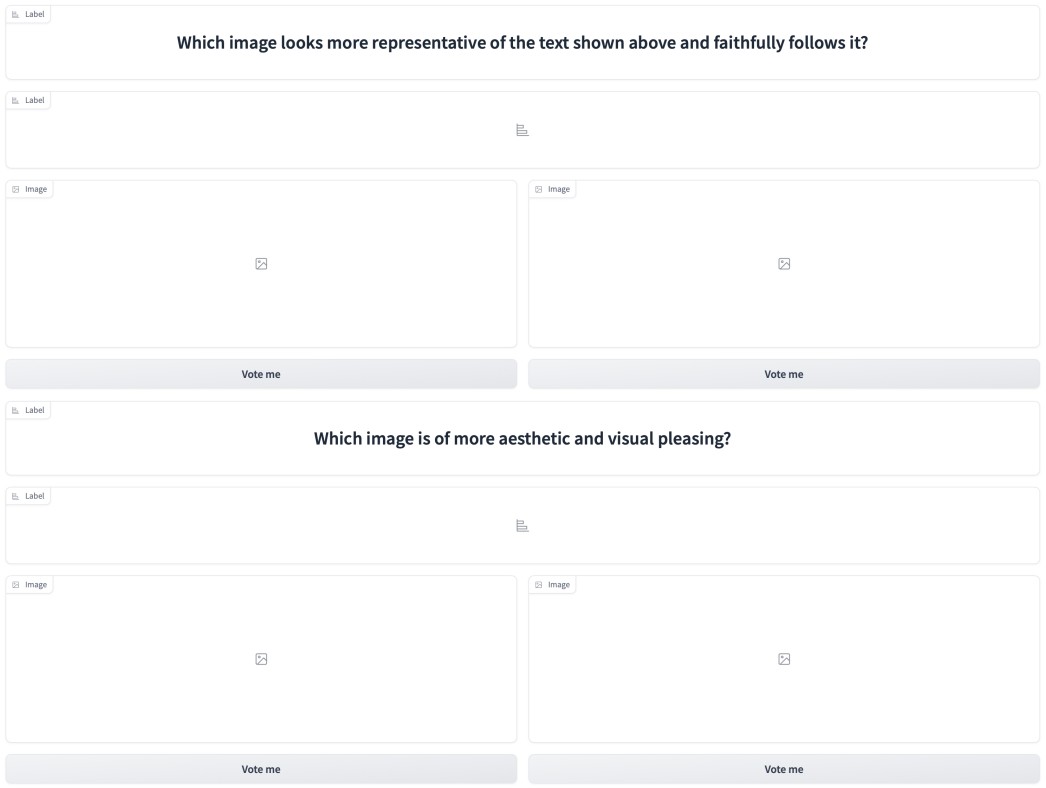

Figure 14: A sample interface for our user preference study, where images are presented in a random left/right order.

## M  Prompts for Figure 1, Figure 2, and Figure 12

We use the following prompts for Figure 1. From left to right, top to bottom:

- a girl examining an ammonite fossil
- A photo of an astronaut riding a horse in the forest.
- a giant gorilla at the top of the Empire State Building
- A close-up photo of a wombat wearing a red backpack and raising both arms in the air. Mount Rushmore is in the background.
- An oil painting of two rabbits in the style of American Gothic, wearing the same clothes as in the original.
- a portrait of an old man
- a watermelon chair
- A sloth in a go kart on a race track. The sloth is holding a banana in one hand. There is a banana peel on the track in the background.
- a penguin standing on a sidewalk
- a teddy bear on a skateboard in times square

We use the following prompts for Figure 2. From left to right, top to bottom:

- a chimpanzee sitting on a wooden bench
- a cat reading a newspaper
- A television made of water that displays an image of a cityscape at night.
- a portrait of a statue of the Egyptian god Anubis wearing aviator goggles, white t-shirt and leather jacket. The city of Los Angeles is in the background.
- a squirrell driving a toy car
- an elephant walking on the Great Wall
- a capybara made of voxels sitting in a field
- Cinematic photo of a beautiful girl riding a dinosaur in a jungle with mud, sunny day shiny clear sky. 35mm photograph, film, professional, 4k, highly detailed.
- A still image of a humanoid cat posing with a hat and jacket in a bar.
- A soft beam of light shines down on an armored granite wombat warrior statue holding a broad sword. The statue stands an ornate pedestal in the cella of a temple. wide-angle lens. anime oil painting.
- children
- A photograph of the inside of a subway train. There are red pandas sitting on the seats. One of them is reading a newspaper. The window shows the jungle in the background.
- a goat wearing headphones
- motion
- A close-up of a woman's face, lit by the soft glow of a neon sign in a dimly lit, retro diner, hinting at a narrative of longing and nostalgia.

We use the following prompts for Figure 12. From left to right, top to bottom:

- A close-up of a woman's face, lit by the soft glow of a neon sign in a dimly lit, retro diner, hinting at a narrative of longing and nostalgia.
- a cat reading a newspaper
- A television made of water that displays an image of a cityscape at night.
- a portrait of a statue of the Egyptian god Anubis wearing aviator goggles, white t-shirt and leather jacket. The city of Los Angeles is in the background.
- a squirrell driving a toy car
- an elephant walking on the Great Wall
- a capybara made of voxels sitting in a field
- A soft beam of light shines down on an armored granite wombat warrior statue holding a broad sword. The statue stands an ornate pedestal in the cella of a temple. wide-angle lens. anime oil painting.
- a goat wearing headphones
- An oil painting of two rabbits in the style of American Gothic, wearing the same clothes as in the original.
- a girl examining an ammonite fossil
- a chimpanzee sitting on a wooden bench

- children
- A still image of a humanoid cat posing with a hat and jacket in a bar.
- motion

