# OpenReview forum: "Improved Distribution Matching Distillation for Fast Image Synthesis"
_NeurIPS.cc/2024/Conference — NeurIPS 2024 oral_

### Official Review · Reviewer_xaYh · 2024-07-06

**Soundness:** 4
**Presentation:** 3
**Contribution:** 4
**Rating:** 8
**Confidence:** 4

**Summary:**

The authors propose an improved way of distilling image-generating diffusion models into fast models capable of generating high-quality images with as few as 1-4 steps. Compared to prior work using distribution-matching (DMD), they do away with the regression loss term that tied the teacher path to the student path. They also introduce a GAN-style loss in the pipeline, and introduce a few other tricks to squeeze out extra performance. They demonstrate SOTA performance among efficient models, and even beat the teacher model (made possible by training using real images and a GAN loss).

**Strengths:**

Overall, I found this to be an excellent paper. The work is well-motivated, addressing an important problem, and will likely be interesting to a large audience. The writing is excellent, with all concepts being well explained. The experimental coverage is excellent, with evaluation on two datasets and with a user study, leaving nothing more to be desired. Their performance numbers are also convincing. Finally, their attention to detail on experimental parameters also looks very thorough, giving confidence that their results can be reproduced.

On a more detailed level, I personally read the original DMD paper not too long ago, and found their regression loss to be slightly unsatisfactory, since it ties the student generation paths to the teacher paths in a way that seems contrary to the idea of the distribution matching loss. Therefore, I was happy to see in this paper that one can do away with this term.

**Weaknesses:**

There's not so much to say here, since I found most aspects of the paper to be excellent. However, I would have liked to see some more details in how their approach relates to competing approaches. Most notably, the line of work by Sauer et al [23, 24] use SDS and a GAN-style loss. Now that a GAN-style loss is introduced in the DMD framework, the gap between these two approaches gets smaller, and it would be nice to see some more explanation about what the core difference is.

The progress is mostly empirical in nature. For example, the authors note that the training becomes more stable using the two-scale update rule, but they don't present any theoretical convergence guarantees (which is perfectly fine, the empirical progress is definitely good enough in my opinion).

There are a few minor typos (e.g. line 122, should be "gradient of the data log likelihood"), but few enough not to impair the overall understanding (and I trust the authors to do a final proof reading for the camera ready version).

**Questions:**

What is the most significant difference between this work and the line of work by Sauer et al ([23, 24])? Could one sentence about that be added to the "related work" section?

The original DMD paper [22] demonstrated examples of mode collapse when omitting the regression loss. GANs (at least some) are also known to be prone to mode collapse. Could you mention anything about the mode collapse situation in this method? (Since the FID numbers are good, I assume that this is also good, but since this was a main point of analysis in [22]?)

**Limitations:**

Yes, limitation and potential social impact are well-described in section 6 and A.

---

> ### Author Rebuttal · Authors · 2024-08-06
>
> We sincerely appreciate Reviewer xaYh's constructive feedback. We will fix all typos. Below, we address the remaining concerns.
>
> **How is DMD2 related to ADD and LADD? What is the most significant difference?**
>
> Thank you for the opportunity to discuss how DMD2 relates to other concurrent GAN based methods such as ADD and LADD. The ADD paper [23] utilizes a pretrained DINO-based classifier in pixel space, which we found to be less efficient in terms of training and leading to reduced diversity, as detailed in Table 6. More recent efforts like LADD, UFOGEN, and SDXL-Lightning employ latent diffusion GAN-based networks akin to our DMD2. A significant difference, which is often overlooked, is that DMD-style training inherently integrates classifier-free guidance directly into the real score of the DMD gradient. This integration simplifies the training process of our model. In contrast, purely GAN-based methods typically struggle to incorporate classifier-free guidance directly. For example, SDXL-Lightning needs to combine GAN with progressive distillation to enable CFG, while LADD relies on diffusion generated images with CFG as real data in its GAN discriminator. These approaches often complicate the training process. We will elaborate further on these distinctions in the related work section of our revised paper.
>
> **The original DMD paper [22] demonstrated examples of mode collapse when omitting the regression loss. GANs (at least some) are also known to be prone to mode collapse. Could you mention anything about the mode collapse situation in this method? (Since the FID numbers are good, I assume that this is also good, but since this was a main point of analysis in [22]?)**
>
> Similar to our response to Reviewer xYYF, we assess the mode collapse situation under two scenarios:
>
> - Class-Conditional Image Generation: We observed no mode collapse, as evidenced by the state-of-the-art FID scores for image generation (see Table 1).
> - Text-to-Image Generation: This scenario is more complex. We utilize classifier-free guidance, which typically trades diversity for image quality. At high guidance settings, using the SDXL baseline, we achieved superior image quality with diversity comparable to or better than other distillation methods but slightly worse than the teacher model ((see Table 6 and the new results in our Response to Reviewer aF5v).
>
> Our current framework trains stably and produces excellent image quality. However, there remains a small gap in output diversity compared to the original teacher model and we are open to exploring future methods that might enhance this diversity/quality tradeoff by better integrating trajectory-preserving techniques (such as the more efficient consistency distillation) with distribution matching methods. We believe this is a promising direction for future research.

---

> > ### Comment · Reviewer_xaYh · 2024-08-08
> >
> > I read the rebuttal and thank the authors for good responses to my questions. I have no further comments, and my "strong accept" recommendation stands. I wish the authors good luck, and I'm looking forward to read the final version!

---

### Official Review · Reviewer_uheV · 2024-07-08

**Soundness:** 3
**Presentation:** 3
**Contribution:** 3
**Rating:** 6
**Confidence:** 4

**Summary:**

This work proposed an improved training method for distribution matching distillation, named DMD2. Notably, compared to DMD, it does need the regression loss which relies on constructing the synthetic noise-data pairs. Instead, DMD2 introduces three new features: 1) a two time-scale update rule for fake score and generator training, 2) a GAN loss which extracts the bottleneck features of fake score network for a training prediction head as the discriminator, and 3) backward simulation for few-step distillation that produces inference-time generator inputs during training.

**Strengths:**

- The paper is well written and easy to read.
- The new distribution matching distillation technique (DMD2) significantly improves over DMD with several well-justified innovations, such as TTUR, GAN loss and backward simulation.
- DMD2 achieves SOTA performance on ImageNet-64 and COCO 2014 with one-step generation, and can achieve SOTA performance in distilling SDXL to a 4-step generator, measured by sufficient automatic metrics and human studies.
- Ablation studies have been well executed to highlight the importance of each introduced feature.

**Weaknesses:**

- The proposed method introduces many hyperparameters and seems to be sensitive to these hyperparameters, such as batch size, guidance scale for teacher score, GAN loss weighting and fake score updating frequency. For instance, in different distillation tasks (EDM on ImageNet, SDv1.5 and SDXL), these hyperparameters are different (as shown in Appendix G). I’m not sure if the hyperparameters need to be specifically tuned for good performance in each distillation task.
- The authors claimed that “SDXL remains challenging to distill into a one-step generator because of limited model capacity and a complex optimization landscape to the direct mapping from noise to highly diverse and detailed images”. For the point of “limited model capacity”, does it mean that if the student has the same network with SDXL, by any means, we are not able to achieve a one-step generation that matches SDXL’s performance? For the point of “a complex optimization landscape”, it seems that both SDv1.5 and SDXL are trained on LAION dataset, which means they are both trying to learn the same mapping from noise to data. Does it mean the higher-quality generation of SDXL (rather the training data of teacher models) hinders the one-step distillation? On the other hand, I wonder if it is possible to tune the hyperparameters of distilling SDXL for a better one-step generator. For example, in Appendix G, the batch size for distilling SDXL is only 128 while the batch size for distilling SDv1.5 is only 2048. If DMD2 is sensitive to batch size in the large-scale text-to-image case, can we increase the batch size for distilling SDXL to 2048 during training for improved performance?
- There are some inconsistencies: 1) In Figure 4, the caption says the teacher uses 50 sampling steps while the main text says “while requiring 25x fewer forward passes (4 vs 100)”. 2) EDM (Teacher ODE) originally reported their FID as 2.22, while this work reports 2.32 in Table 1. 3) Both $\mu_{\text{real}}$ and $\mu_{\text{fake}}$ are introduced without definition.

**Questions:**

- I’m curious about the extra cost of backward simulation, i.e., producing synthetic images with the current student generator running several steps. Is it possible to compare the training time per iteration with and without backward simulation?
- From the numbers in Table 2, it looks like the 4-step distillation improves Patch FID but gets worse FID and CLIP, compared to 1-step distillation. Any justification?

**Limitations:**

The authors adequately addressed the limitations and potential negative societal impact of their work.

---

> ### Author Rebuttal · Authors · 2024-08-06
>
> We sincerely appreciate Reviewer uheV's constructive feedback. Below, we address the remaining concerns.
>
> **How to select the set of hyperparameters?**
>
> Thank you for your question. Our approach to selecting hyperparameters is straightforward and consistent across all datasets. We utilize the maximum batch size our compute setup allows. The learning rate is determined as the highest value that does not cause divergent loss within the first 500 iterations. We set the number of TTUR iterations to the minimum required for stability. The guidance scale is chosen based on what yields the best results for the teacher model. For the GAN weight, our method works well across a wide range of values, as demonstrated by the following ImageNet FID scores:
>
> | Weight  | ImaegNet FID |
> | - | - |
> | 2e-3 | 1.31 |
> | 3e-3 | 1.28 |
> | 4e-3 | 1.28 |
> | 5e-3 | 1.26 |
> | 1e-2 | 1.30 |
>
> We will include these guidelines in our updated version.
>
> **For the point of “limited model capacity”, does it mean that if the student has the same network with SDXL, by any means, we are not able to achieve a one-step generation that matches SDXL’s performance?**
>
> We are open to the possibility of a one-step generation that matches the performance of the teacher model. However, we want to emphasize that achieving this in a single-step generation is substantially more challenging than in a multi-step process, especially when distilling complex networks like SDXL. While our one-step generator can match the teacher model in quantitative metrics such as FID, qualitative aspects of image quality present greater challenges. Visually, the one-step process still produces some artifacts that are difficult to eliminate (See Figure 11). An exciting direction for future research could involve refining our training objectives to address these issues more directly, as seen in recent developments like those explored in HyperSD [1].
>
> **For the point of “a complex optimization landscape”, it seems that both SDv1.5 and SDXL are trained on LAION dataset, which means they are both trying to learn the same mapping from noise to data. Does it mean the higher-quality generation of SDXL (rather the training data of teacher models) hinders the one-step distillation?**
>
> While both models are trained on the LAION dataset, they likely utilize different subsets, leading to variations in the noise-to-data mapping. The higher quality and particularly the subtler details found in SDXL are indeed more challenging to capture, potentially requiring more advancements in loss design.
>
>  **I wonder if it is possible to tune the hyperparameters of distilling SDXL for a better one-step generator. For example, in Appendix G, the batch size for distilling SDXL is only 128 while the batch size for distilling SDv1.5 is only 2048. If DMD2 is sensitive to batch size in the large-scale text-to-image case, can we increase the batch size for distilling SDXL to 2048 during training for improved performance?**
>
> We have observed improved performance with increased batch sizes and compute. We used a batch size of 128 for SDXL, as this is the maximum our current setup can accommodate within our compute budget of 3 days—especially considering that the SDXL model is three times larger than SDv1.5. Exploring a larger batch size, such as 2048, with additional resources in the future would indeed be an interesting experiment to potentially enhance performance further.
>
> **Some Writing Inconsistency**
>
> Thank you for pointing out these inconsistencies. Regarding the discrepancy between 50 and 100 steps in Figure 4, the teacher model uses 50 steps, but effectively has 100 forward passes due to the application of classifier-free guidance. For EDM, upon reevaluating the released model, we recorded a FID of 2.32, which we will update to 2.22 in our revised version to reflect the most accurate data. We appreciate your attention to detail and will correct the remaining issues. Thank you again for your valuable feedback.
>
> **Extra cost per training iteration of backward simulation**
>
> Thank you for raising the issue of training computational cost. The training iteration time for the model with backward simulation is approximately 9.2 seconds per iteration, compared to 7.8 seconds for the model without backward simulation. We also discovered that enabling backward simulation only during the last 20% of the training epochs yields comparable results. This approach offers a more computationally efficient option when resources are limited.
>
> **From the numbers in Table 2, it looks like the 4-step distillation improves Patch FID but gets worse FID and CLIP, compared to 1-step distillation. Any justification?**
>
> From the data in Table 2, the difference in FID between the one-step and four-step distillation (0.3) is generally within the range of variability observed across different runs, indicating comparable performance. The significant improvement in Patch FID for the four-step distillation reflects better local image detail, aligning with qualitative improvements we observed. However, the CLIP score did decrease, which may suggest that the four-step method slightly sacrifices prompt alignment for image quality. This also happens for previous approaches with good text to image alignment like SDXL-Turbo.
>
> [1] Ren, Yuxi, et al. "Hyper-sd: Trajectory segmented consistency model for efficient image synthesis." arXiv preprint arXiv:2404.13686 (2024).

---

### Official Review · Reviewer_xYYF · 2024-07-09

**Soundness:** 3
**Presentation:** 3
**Contribution:** 3
**Rating:** 6
**Confidence:** 4

**Summary:**

This paper introduces DMD2, a few-step distilled generator to achieve fast sampling while maintaining the decent generation quality of the multi-step diffusion models. DMD2 proposes several new improvements to the training procedure of the original DMD, including (1) replacing the regression loss with the Two-Time scale Update Rule (TTUR) to stabilize the training process, (2) incorporating the standard GAN loss to achieve better quality, (3) and utilizing the backward simulation to alleviate the potential mismatch of training and inference. Built upon these modifications, DMD2 achieves excellent results on few-step image generation.

**Strengths:**

1. The source-intensive process of generating noise-image pairs is replaced by a simple TTUR strategy.
2. DMD2 achieves SOTA results on one-step image generation on ImageNet64 and shows its effectiveness on distilling SDXL into few steps.

**Weaknesses:**

1. It would be better to include a detailed training algorithm to clearly showcase the modifications over the original DMD training process.
2. In practice, the real data used to train the (teacher) diffusion models may not be accessible to the users who hope to distill a small (student) generation model (due to privacy, storage, …). In that case, one limitation of DMD2 is the calculation of GAN loss may become inapplicable. Could you share your opinions about this matter?

**Questions:**

1. In the original DMD paper, the regression loss is capable of mitigating the issue of mode collapse. Would DMD2 also suffer from this issue as the regression loss is removed and an extra GAN loss is introduced?
2. Why the intermediate outputs of DMD2 shown in the right subfigure of Figure 3 are so similar and seem to follow a certain trajectory? As far as I know, distribution matching-based methods do not guarantee the specific paths of the teacher diffusion model and the student generation model are aligned (as mentioned in Lines 35-39). Could authors provide more explanations about this phenomenon (Fig 3)? Note that the samples generated by few-step consistency models may also switch between different paths and generate very different images.

**Limitations:**

Please refer to the weakness and question sections above.

---

> ### Author Rebuttal · Authors · 2024-08-06
>
> We sincerely appreciate Reviewer aF5v's constructive feedback. Below, we address the remaining concerns.
>
> **It would be better to include a detailed training algorithm to clearly showcase the modifications over the original DMD training process.**
>
> Thank you for your suggestion. As you accurately summarized, our DMD2 model eliminates the need for the regression loss and incorporates a diffusion GAN loss along with backward simulation to mitigate training-inference mismatches. To clearly illustrate these modifications over the original DMD training process, we will include a detailed comparison of the training algorithms in the revised version of our paper.
>
> **In practice, the real data used to train the (teacher) diffusion models may not be accessible to the users who hope to distill a small (student) generation model (due to privacy, storage, …). In that case, one limitation of DMD2 is the calculation of GAN loss may become inapplicable. Could you share your opinions about this matter?**
>
> We acknowledge that DMD2 relies on access to real data to enhance diversity and image quality. However, it is important to note that the exact dataset used to train the teacher model is not required. For our training, we utilized a random set of 500,000 images from the LAION database, which is generally of lower quality than the curated aesthetic dataset of SDXL. This demonstrates that our GAN loss can be effectively applied using an alternative dataset, underscoring its versatility and universal applicability, regardless of the specific model being distilled.
>
>
> **In the original DMD paper, the regression loss is capable of mitigating the issue of mode collapse. Would DMD2 also suffer from this issue as the regression loss is removed and an extra GAN loss is introduced?**
>
> Thank you for your question. In DMD2, we demonstrate that much of the mode collapse observed in the original DMD method relates more to training issues than to an inherent inability of the DMD loss to support diverse generator training. Practically, we can assess the final performance under two scenarios:
>
> - Class-Conditional Image Generation: We observed no mode collapse, as evidenced by the state-of-the-art FID scores for image generation (see Table 1).
> - Text-to-Image Generation: This scenario is more complex. We utilize classifier-free guidance, which typically trades diversity for image quality. At high guidance settings, using the SDXL baseline, we achieved superior image quality with diversity comparable to or better than other distillation methods but slightly worse than the teacher model (see Table 6 and the new results in our Response to Reviewer **aF5v**).
>
> Our current framework trains stably and produces excellent image quality along with comparable diversity. However, we are open to exploring future methods that might enhance this diversity/quality tradeoff by better integrating trajectory-preserving techniques (such as the more efficient consistency distillation) with distribution matching methods. We believe this is a promising direction for further research.
>
> **Why intermediate outputs of DMD2 follow a certain trajectory?**
>
> This observation was initially surprising to us as well. Although our generator does not follow the teacher diffusion model’s sampling trajectory, our training methods lead to few-step generators where the first output significantly shapes the general structure. Subsequent images tend to closely resemble this initial output. This effect is likely influenced by the relatively high signal-to-noise ratios in subsequent sampling steps, which preserve much of the structure even after noise injection.

---

### Official Review · Reviewer_aF5v · 2024-07-12

**Soundness:** 4
**Presentation:** 3
**Contribution:** 3
**Rating:** 8
**Confidence:** 5

**Summary:**

This work addresses identifies reasons for training instability of one of competitive diffusion distillation approaches based on distribution matching, using bi-level optimization and also adopt a GAN based feature space feedback for improved quality. Overall demonstrate very good performance on SDXL, SD checkpoints demonstrating effectiveness on large models too.

**Strengths:**

Paper is well written and easy to understand, with good benchmarking for large scale models and also comparing to other distillation techniques.  Overall, DMD puts lesser constraints on distillation w.r.t underlying map from noise to data space enabling more good formulation for distillation. And improving stability is useful for broader adoption of DMD style formulation for distillation towards practical diffusion based applications.

**Weaknesses:**

As the authors discuss on not using real-data within current formulation and setup, there could be a tendency for model to have model collapse?

As proposed distillation objective is not sampling w.r.t teacher's marginal predictive distribution nor data-distribution. It would be useful to get some diversity metric at per-prompt level using  e.g., LPIPS Diversity or etc comparing to other Distillation approaches and teacher model.

Also it would be useful to understand what stage of training causes this mode collapse, i.e., does small scale training preserve diversity at cost of some quality drop or we observe a consistent drop in diversity?

**Questions:**

What is setup of ablation without backward simulation in multi-step setting? Do forward diffusion and query the student generator and based on that query fake score function estimator? If so is the fake score function also trained on equivalent forward diffuse + generator's predictive distribution? As it is currently unclear is it alignment of fake score function alignment to generator or the backward simulation which is resulting in improved performance.

Also given DMD has implicit assumption that the fake score function is capturing predictive distribution of student generator and authors identify its fitting being one of reasons for instability. It might be useful to understand how sensitive is alignment of fake score estimation to student's generator. Also, how good is quality of fake score fun at different stages of training and its implications?

As at implementation level we are starting with pretrained model and in the limit if distilled student model matches pre-trained model's weights we are asking fake score function to match pre-trained model again.

**Limitations:**

This work build on DMD and improves stability, good engineering practice as primary contribution of this work with limited formulation novelty or insights. So more insights on hyperparameter sensitivity, why and how are different design choices effect final performance, diversity etc as discussed above would make it more useful for community and strong contribution!

---

> ### Author Rebuttal · Authors · 2024-08-06
>
> We sincerely appreciate Reviewer aF5v's constructive feedback. Below, we address the remaining concerns.
>
> **As the authors discuss on not using real-data within current formulation and setup, there could be a tendency for model to have model collapse?**
>
> There may be some misunderstanding regarding our use of real data. In fact, our model does incorporate real data during training via the GAN loss component. This inclusion enhances diversity and helps to prevent mode collapse. The positive impact of integrating GAN loss with real data is evident in the improved performance detailed in Tables 3 and 4.
>
> **As proposed distillation objective is not sampling w.r.t teacher's marginal predictive distribution nor data-distribution. It would be useful to get some diversity metric at per-prompt level using e.g., LPIPS Diversity or etc comparing to other Distillation approaches and teacher model.**
>
> Thank you for your suggestion! We have indeed conducted a diversity assessment using LPIPS diversity metrics, which is presented in Table 6 of the appendix. Our results indicate that our model's diversity is on par with other distillation approaches, such as the latent consistency model, and significantly exceeds that of purely GAN-based methods like SDXL-Turbo. While our model shows slightly worse diversity compared to the teacher model, it offers substantially better image quality, as illustrated in Figure 5. We believe this often represents a more advantageous trade-off for practical text-to-image applications.
>
> **Also it would be useful to understand what stage of training causes this mode collapse, i.e., does small scale training preserve diversity at cost of some quality drop or we observe a consistent drop in diversity?**
>
> Thank you for your suggestion! In response, we retrained our SDXL model and monitored diversity metrics throughout the training process. The results from this new run showed a small diversity improvement over those reported in our paper. Specifically, the model displays higher diversity at the beginning, which gradually diminishes as image quality improves. Despite this trend, the final diversity scores of the model remain closely comparable to those of the teacher model (0.63 vs 0.64 for the teacher), indicating a well-maintained balance between diversity and image quality.
>
> | Train Iter | LPIPS Diversity |
> | - | - |
> | 0 | 0.25 |
> | 4k | 0.65 |
> | 8k | 0.64 |
> | 12k | 0.65 |
> | 16k | 0.64 |
> | 20k | 0.63 |
> | SD Teacher Baseline | 0.64 |
>
>
>
> **Setting without backward simulation**
>
> The reviewer’s interpretation is correct. In the setting without backward simulation, we add noise to real images and then feed these noisy images into our generator. The output from the generator is then supervised using both the DMD and GAN loss. Additionally, the fake score function is trained based on this generator output. We will clarify this process more thoroughly in the revised version of our paper.
>
> **how sensitive is alignment of fake score estimation to student's generator and how good is the quality of fake score at different stages of training and its implications?**
>
> Thank you for suggesting this set of further analyses. Due to time constraints, these will be included in the revised version of our paper. Regarding the first part, we believe achieving proper alignment between the fake score estimation and the generator’s output distribution is crucial. As shown in Figure 9 of the main paper, more frequent training of the fake score—aimed at enhancing its accuracy—leads to more stable generator training and overall improved performance. We will provide a more comprehensive analysis in the revised version. For the second part, we plan to retrain an ImageNet model and monitor the performance of the fake score model at various training stages by assessing the quality of its sample outputs. We appreciate the reviewers’ valuable suggestions and look forward to incorporating these insights!

---

> > ### Comment · Reviewer_aF5v · 2024-08-08
> > **Thank you for addressing/clarifying most of concerns.**
> >
> > Happy to raise my score.
> >
> > One thing i want to understand from authors, is how stable is this formulation when number of steps go from 20K to say 60K or more? Do you see performance peak around 20K or further training improves quality? unlike original DMD as authors report only 20K was curious if there were any interesting empirical findings which would provide further insights to community.
> >
> > This goes back to fake score function alignment, does it help to reinitialize fake score function with teacher model again as student converges well to teacher? As it seems a bit unclear on properties/fit of fake score function and how it effects overall training stability.
> >
> > Looking forward for more results in later version of paper.

---

> > > ### Author Response · Authors · 2024-08-08
> > >
> > > We are pleased that our responses have addressed your concerns and appreciate your consideration to raise the score!
> > >
> > > Regarding your further inquiries, we currently utilize 20K iterations because this represents almost the maximum compute we can afford (64 GPUs over 3 days). However, we have not yet observed the peak performance of our models. For example, in a trial where we extended to 30K iterations, we managed to improve the FID from 19.3 to 18.7. We are eager to explore extending the training duration in our revised version to further confirm the model's stability.
> > >
> > > The suggestion to reinitialize the fake diffusion model at a later stage is intriguing, and we look forward to experimenting with this approach. Thank you once again for your invaluable suggestions and insights!

---

### Official Review · Reviewer_Yut7 · 2024-07-14

**Soundness:** 4
**Presentation:** 4
**Contribution:** 4
**Rating:** 8
**Confidence:** 4

**Summary:**

The paper introduces an upgraded version of Distribution Matching Distillation (DMD), i.e., DMD2, which addresses the limitation and inefficiency of previous DMD and improves the performance of efficient and high-quality image synthesis using diffusion models. Specifically, the authors identify the limitations of the original Distribution Matching Distillation (DMD), such as the need for a regression loss and extensive dataset construction. DMD2 eliminates the regression loss, integrates a Generative Adversarial Network (GAN) loss, and introduces a two-time-scale update rule to stabilize training. Additionally, a new training procedure is implemented to simulate multi-step sampling, addressing the training-inference mismatch. Experimental results demonstrate that DMD2 achieves state-of-the-art performance, surpassing the original DMD and other competitive models in image quality and efficiency.

**Strengths:**

1. **Elimination of Regression Loss**: By removing the regression loss, DMD2 simplifies the training process and reduces computational costs, making it more scalable and flexible for large-scale applications.

2. **Integration of GAN Loss**: The incorporation of a GAN loss improves the quality of generated images by discriminating between real and generated samples, enhancing the overall distribution matching objective.

3. **Two-Time-Scale Update Rule**: This technique addresses training instability issues, ensuring that the fake score accurately tracks the generator’s output distribution, leading to stable and high-quality image generation.

4. **Multi-Step Sampling**: The introduction of multi-step sampling allows DMD2 to produce high-quality images in fewer steps, addressing the inefficiency of one-step generation while maintaining performance.

5. **Comprehensive Evaluation**: The paper provides extensive experimental results on various benchmarks, demonstrating DMD2's superior performance in both class-conditional and text-to-image synthesis tasks.

**Weaknesses:**

I do not find a specific weakness of this paper.

**Questions:**

- Training with GAN often entails numerical instability. Does DMD2 have such concerns? If it is true, could the author provide some details in overcoming the instability of DMD2?
- Besides the evaluation metric such as FID and Inception scores, how about some human-related metrics such as ImageReward or aesthetic scores? Does DMD2 show comparable results to teacher SDXL?

**Limitations:**

The paper includes limitations of the proposed method, e.g., requires multiple steps to generate on par with teacher model, e.g., SDXL.

---

> ### Author Rebuttal · Authors · 2024-08-06
>
> We sincerely appreciate Reviewer Yut7's constructive feedback. Below, we address the remaining questions regarding numerical instability and human-related metrics.
>
> **Training with GAN often entails numerical instability. Does DMD2 have such concerns? If it is true, could the author provide some details in overcoming the instability of DMD2?**
>
> The GAN component in DMD2 does not introduce numerical instability. This stability can be attributed to our use of a diffusion GAN framework, where both the generators and discriminators are initialized from a pretrained diffusion model. Before classification, images are also treated with noise injection, enhancing stability. This method has proven more stable than traditional GANs that utilize pixel space discriminators without noise injection, as supported by several recent studies [1, 2]. Furthermore, DMD2 includes a two-timestep-scale update rule that bolsters stability. Additionally, DMD2 utilizes a weighted combination of DMD and GAN losses. The DMD loss consistently guides the overall structure of the images, preventing mode collapse and ensuring training stability even with the added GAN components.
>
> **Besides the evaluation metric such as FID and Inception scores, how about some human-related metrics such as ImageReward or aesthetic scores? Does DMD2 show comparable results to teacher SDXL?**
>
> Thank you for this insightful suggestion! We have extended our evaluation of DMD2 and SDXL to include both ImageReward and aesthetic scores, using PartiPrompts [3] for consistency with our human evaluation. Below is a summary of our findings:
>
> | Method | ImageReward | Aesthetic Score |
> | - | - | - |
> |DMD2 | 1.07 | 6.30 |
> | SDXL | 0.86 | 6.16 |
>
> These results demonstrate that DMD2 consistently outperforms SDXL, corroborating the findings from our main paper's FID and human evaluation results. We will incorporate these additional metrics into the revised version of our paper.
>
> [1] Wang, Zhendong, et al. "Diffusion-gan: Training gans with diffusion." ICLR 2023
>
> [2] Xu, Yanwu, et al. "Ufogen: You forward once large scale text-to-image generation via diffusion gans." CVPR 2024.
>
> [3] Yu, Jiahui, et al. "Scaling autoregressive models for content-rich text-to-image generation." arXiv preprint arXiv:2206.10789 2.3 (2022)

---

> > ### Comment · Reviewer_Yut7 · 2024-08-11
> >
> > Thank you for the rebuttal. I will maintain my original score.

---

### Author Rebuttal · Authors · 2024-08-06

We sincerely thank all reviewers for their constructive feedback. We are grateful for the positive reception of our work, which has been recognized for its well-founded innovations and outstanding quality. Our DMD2 model facilitates the training of a few-step generator that delivers superior image quality and diversity comparable to the teacher SDXL models. We have incorporated additional evaluations concerning human-related and diversity metrics as requested by Reviewer **Yut7** and Reviewer **aF5v**. Detailed responses to each reviewer’s specific concerns are provided below.

---

### Decision · Program_Chairs · 2024-09-25

**Decision:**

Accept (oral)

**Comment:**

This paper presents DMD2, a few-step distilled generator designed to achieve fast sampling while preserving the high-generation quality of multi-step diffusion models. The paper introduces several key enhancements to the original DMD's training procedure, including: (1) replacing the regression loss with the Two-Time Scale Update Rule to stabilize training, (2) integrating GAN loss to improve generation quality, and (3) employing backward simulation to address potential mismatches between training and inference. Experimental results show that DMD2 achieves state-of-the-art performance, outperforming the original DMD and other competitive models in both image quality and efficiency.

All reviewers unanimously acknowledged the novelty and effectiveness of this work, leading to its acceptance. The authors should also revise their paper according to the reviewers' suggestions in the final version.